# Effect of Entrance Frame on Crack Development around Prefabricated Subway Station Openings

**Zhenze Mo [1], Shuaike Feng [2], Dongzhi Guan [2,*] and Zhengxing Guo [2]**

1   Xicheng-CRRC (Wuxi) Urban Rail Transit Engineering Co., Ltd., Wuxi 214432, China
2   School of Civil Engineering, Southeast University, Nanjing 211189, China
*   Correspondence: guandongzhi@seu.edu.cn

**Abstract:** The openings at the sidewalls of subway station entrances generally reduce the localized load-bearing capacity of the sidewalls and lead to concentrated stress around the openings. In this study, to strengthen the sidewalls with openings in a newly-developed prefabricated subway station, a prefabricated steel-reinforced concrete (SRC) frame around the entrance was developed. To further investigate the effect of the developed entrance frame on the mechanical behavior of the sidewalls, a monotonic static test and finite element analysis were performed on a 1/2 scale station entrance substructure, including the proposed entrance frame and the adjacent top slab, bottom slab, and sidewalls. It was found that the developed entrance frame could effectively prevent stress concentration in the adjacent sidewall region. The most severe crack development was concentrated at the corner of the opening, which could be attributed to the torsional moment at the SRC beam end. The ratio of the torque shared by the beam to the total bending moment of the slab end varied from 21.2% to 26.8% in the elastic stage of all cases. In addition, both the improvement in the torsional bearing capacity of the SRC beam and the out-of-plane flexural capacity of the SRC column could positively contribute to controlling the crack development around the opening.

**Keywords:** prefabricated subway stations; monotone static test; entrance frame; development of cracks; stress distribution

## 1. Introduction

A prefabricated subway station that comprises assembled technology applied to the construction of the station conforms to the sustainable construction development trend. Unlike the construction of traditional subway stations, which involves unavoidable carbon emissions and serious resource consumption, assembled station construction technology is not only energy-saving and environmentally friendly but also has advantages such as short construction periods, reliable product quality, and a high degree of industrialization [1,2]. The assembly technology was first researched and developed in the construction of the Changchun subway station in China. The proposed prefabricated subway station structure comprises prefabricated elements such as bottom plates, sidewalls, and top arches. A novel grouted mortise–tenon joint was developed to connect the aforementioned prefabricated components. Furthermore, Yang et al. [3–5] experimentally assessed the bearing capacity characteristics and flexural stiffness of the developed joints and found that the proposed joints could effectively transfer the internal forces of prefabricated elements. Additionally, theoretical formulas were derived to calculate the ultimate flexural resistance and bending stiffness of the novel joints. Tao et al. [1,6] conducted a shaking table test to evaluate the seismic response of a prefabricated station structure and found that the mortise–tenon joint exhibited a higher bearing capacity than that of the prefabricated components, guaranteeing excellent integrity and mechanical properties of the station structure under seismic loads. Ding et al. [7] performed a numerical simulation of the single-ring structure of a Changchun prefabricated subway station and traditional cast-in situ subway station. Comprehensive

analysis indicated that the single-ring structure of the Changchun station exhibited more favorable deformation resistance and mechanical properties than those of the traditional station. The popularization and application of assembly technology in the construction of subway stations have resulted in many novel connected methods and precast element forms being extensively investigated and employed in a variety of station structures. Liu et al. [8] studied the mechanical behavior of a precast concrete beam–slab–column interior joint comprising many connected methods and optimized the thickness of semi-precast slabs in the Jinanqiao prefabricated subway station. Du et al. [9] subsequently presented an experimental program on precast sidewalls with grouted sleeve connectors, which provided a foundation for predicting the mechanical properties of grouted splices used in prefabricated subway stations. Additionally, a novel wall–beam–strut joint connected using welded steel plates was adopted in the Guangzhou subway station, and the bearing capacity of the proposed joint was experimentally investigated [10]. Thus, the connected and assembled forms in subway prefabricated stations were implemented with reference to above-ground prefabricated structures, and the assessment of the mechanical behavior was performed through experimental and numerical investigations.

A subway station entrance is the passageway for to enter and exit the station meant for the evacuation and transfer of passengers. Therefore, it is necessary to know about the structural behavior and the design method of an entrance structure. Wang et al. [11] investigated structural behavior of a subway station entrance structure under three different buried depths. The optimal sectional dimensions of the deep buried entrance structure were proposed according to the analysis results. Monika [12] reported and discussed the measured displacement of the entrance hall to the metro station. Ou [13] compared the analysis method of an entrance structure in a subway station using a 2D plane model and 3D space models, and the results showed that the corners of an entrance structure should be enhanced. Liu et al. [14] analyzed the pressure distribution of the horizontal entrance of a subway station, which was caused by airflow. Yang and Lin [15] introduced a prefabricated ring frame structure of an entrance and exit in a prefabricated underground metro station structure. Choi et al. [16] presented the project-specific design approach of the station structure, which could be followed for the entrance structure. Most investigations on the mechanical performance of entrance structures in subway stations focus on conventional monolithic entrance structures. Studies on the prefabricated entrance structures are lacking, especially in the progress of promoting prefabricated underground stations.

The design of the entrance inevitably requires large openings to be cut into structural sidewalls, which directly reduces the localized load-bearing capacity of the sidewalls, and this should be rigorously estimated [17]. The mechanical performance of sidewalls with openings in subway stations has been investigated less intensively. Previous designers commonly referred to the recommendations provided by current standards for above-ground structures to evaluate the effects of cut-out openings on the behavior of station sidewalls. Standards such as EN 1992-1-1 [18], AS 3600 [19], ACI 318 [20], and CAN/CSA-A23.3 [21] all provide calculation assumptions to evaluate the mechanical behavior of the reinforced concrete (RC) walls with openings. However, as the load pattern applied to underground sidewalls may be different from that applied to above-ground structural walls, the feasibility of the current design recommendations requires further assessment.

RC structural walls in the above-ground structures can be considered as compression members that are designed to carry in-plane vertical loads [22]. Although a small eccentricity may be present, the failure pattern of the wall is still dominated by compressive forces [23–27]. The load applied to RC structural walls can present another type of the in-plane lateral seismic forces. Most previous studies have focused on the seismic performance of structural walls [28–30].

In contrast, sidewalls in subway stations are generally subjected to negligible in-plane lateral loads [31]. They are typically designed to carry in-plane axial loads and lateral out-of-plane loads such as moments from the top slab ends. Note that the out-of-plane behavior of RC walls with openings has also received little attention in above-ground

structures. Hence, extensive experimental tests may be required to evaluate the out-of-plane mechanical performance of sidewalls with openings.

In general, for all subway stations, the area of the entrance openings introduced in the sidewalls is less than a tenth of the total continuous sidewall area. At the structural level, the entrance opening generally does not significantly affect the ultimate strength of integral sidewalls [18,19]. However, entrance openings in the sidewalls inevitably affect the load transfer paths and cause a redistribution of sidewall stresses. This further results in stress concentration around the opening, which encourages the development of cracks around the opening. Compared with above-ground buildings, the crack width of the components in underground structures must be strictly controlled to satisfy waterproofing requirements [32]. In such cases, the vicinity of the opening where stress concentrations and cracks generally occur first should be strengthened [33].

To strengthen RC walls with openings, additional reinforcing bars or steel plates are always embedded around the openings [34], and some advanced composites on the surface of the cut-outs can be used as externally bonded reinforcements [35]. Most previous studies on the effectiveness of reinforcement details around the edges of the openings have focused on above-ground structures. Therefore, the effects of the above reinforcement methods on the static behavior of sidewalls should be re-evaluated under the specific load pattern carried by the subway station. In addition, the application of these methods in prefabricated subway stations has been limited, owing to complex fabrication and high costs. To improve the efficiency of on-site construction, a steel-reinforced concrete (SRC) frame around the entrance was developed for a prefabricated subway station. The developed entrance frame structure can be used to compensate for the effects of entrance openings on the localized bearing capacity of the continuous sidewalls and to prevent stress concentrations around the openings.

Figure 1 shows the connection of the proposed entrance frame and its adjacent members. To facilitate component transportation, the precast column and beam in the SRC frame can be separately manufactured and connected onsite. As illustrated in Figure 1a, H-steels partially embedded into the bottom slab of the entrance story are applied to connect the precast SRC column to the bottom slab. Furthermore, a cantilever H-steel beam is integrated into the precast SRC column during production to connect the end of the exposed H-steel beam. As shown in Figure 1b, partial precast sidewalls are used in the prefabricated subway station. Horizontal U-type rebars are arranged longitudinally along the column overlapped with the transverse reinforcements of the sidewalls. Similarly, vertical U-type rebars protruding from the bottom slab are utilized to overlap the longitudinal rebars of the sidewalls. After the entrance frame and its adjacent precast sidewalls are placed at the designed positions, on-site concrete is poured into the reinforcement overlapping regions and the prepared region of the semi-sidewall. Subsequently, a partial precast top slab is placed on the corbel and connected to the entrance frame and sidewalls. As shown in Figure 1c, U-type rebars are formed by extending the longitudinal rebars of the sidewalls and partial precast top slab. The connection type between the sidewalls and top slab can be also regarded as overlapping with the U-type rebars. Finally, on-site concrete is poured into the upper region of the partial top slab to improve structural integrity, as shown in Figure 1d.

Due to the influence of the proposed entrance frame on the mechanical behavior of its adjacent sidewalls being ambiguous, the study of the mechanical behavior of the proposed frame and its adjacent sidewalls under the common stress state is an important prerequisite for promoting the application of the proposed frame in prefabricated subway stations. To experimentally investigate the overall behavior, cracking behavior, and stress distribution of the frame and adjacent members under vertical static loads, in this study, a 1/2 scale station entrance substructure comprising the developed frame, the adjacent top slab, the bottom slab, and sidewalls was constructed and tested under monotonically increasing vertical loads on the top slab. The effect of the frame on the mechanical behavior of its adjacent sidewalls was analyzed. Additionally, a numerical parametric study was

conducted using the FE model to further investigate the influence of the developed frame with different configuration parameters on the crack development around the opening. A comparative analysis based on experimental and numerical tests revealed the crack development mechanism of the openings at the sidewalls of subway station entrances.

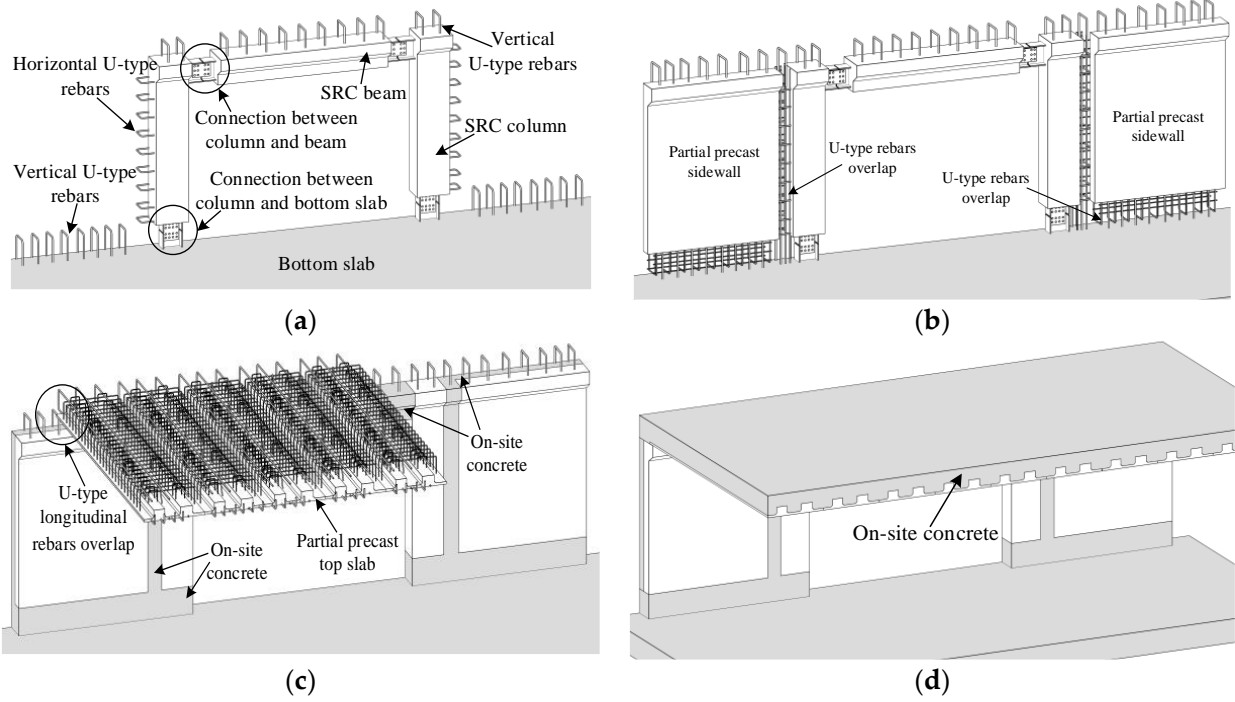

**Figure 1.** Developed entrance system and assembly process: (**a**) entrance frame in assembly, (**b**) sidewalls connected to entrance frame, (**c**) top slab connected to frame and sidewalls, and (**d**) assembled entrance system.

## 2. Experimental Program

### 2.1. Representative Subway Station

An actual engineering case under construction in China was selected as the prototype structure for this study. The subway station comprises a two-story, three-span structure, as shown in Figure 2. The total height of the story where the entrance system is located is 6600 mm, of which 800 mm is the thickness of the laminated top slab, and 5800 mm is the height of the precast sidewalls. The entrance opening dimensions are 4800 (height) × 7000 (width) mm, and the thickness of the laminated sidewalls is 700 mm, wherein the cast-in-place and precast layers are 350 mm. The height of the entrance frame is 5800 mm, and the distance between the column centers is 7800 mm. The cross-sectional dimensions of the columns are 800 × 700 mm, and the height and width of the beam are 1000 and 700 mm, respectively. Furthermore, the thickness of the entrance frame is identical to that of the adjacent sidewalls. One entrance substructure (Figure 2) is taken out from the top floor of the prototype building. The total height of the substructure was uniquely determined by the height of the story wherein the entrance system was located. The overhang length of the assembled top slab was determined based on the bending moment diagram of the top story of a subway station under uniform vertical loads (Figure 3). A single standardized precast sidewall panel was retained on both sides of the entrance frame, and the width of each sidewall panel was 3000 mm. In addition, the boundary conditions of the substructure are completely consistent with the actual station structure.

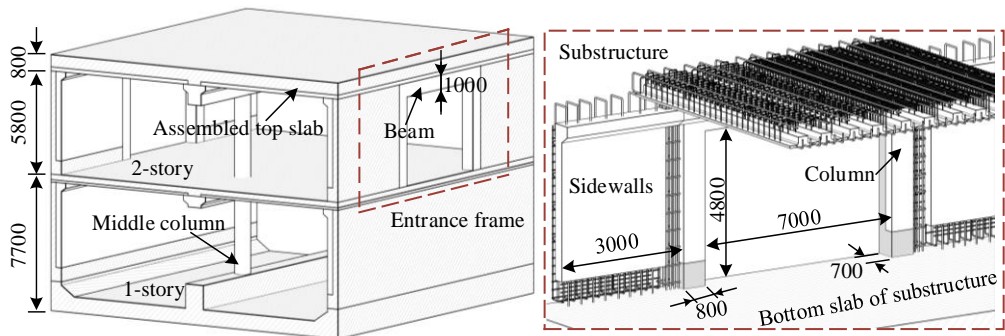

**Figure 2.** Schematic illustration of the full-scale subway station (units: mm).

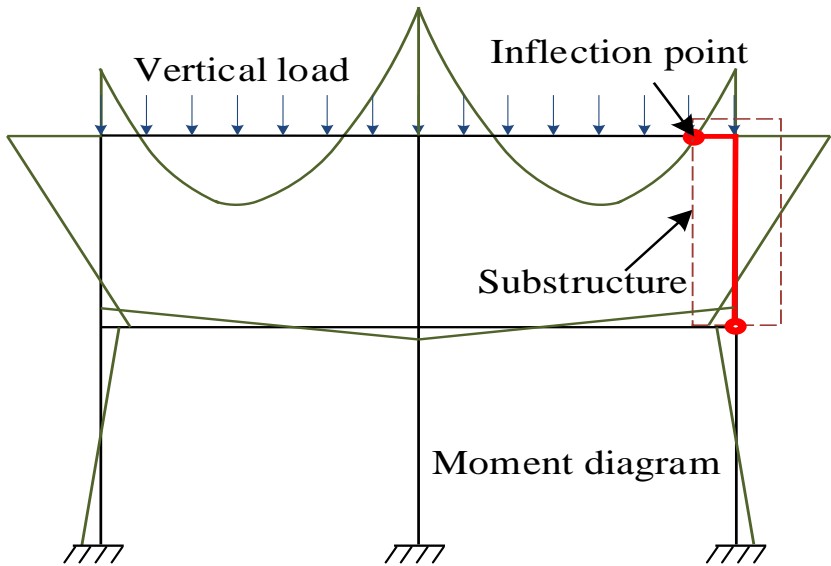

**Figure 3.** Simplified model of a cross-section of the subway station.

### 2.2. Specimen Details

As the constraints of the available experimental manipulation space, the substructure was scaled down by a factor of 2 to serve as the test specimen. Table 1 summarizes the parameters of the elements in the specimen. Scaled-down specimens are commonly used for studying the effects of cut-out openings on the strength of RC walls [17,22,26]. Previous studies have determined that it is feasible to test scaled-down RC walls with cut-out openings to further understand the mechanical properties of the structures. The design of the substructure considered the similarity between the scaled-down specimen and the original structure. The ratio of the geometric dimensions of each member in the substructure model to those of the corresponding member in the actual station structure was maintained at 1/2, and the substructure model was completely consistent with the material used in the actual station structure. Therefore, according to the scalability theory for the static loading test on a structure of civil engineering, the load-bearing mechanism and force-transferring path of the test scale model can be regarded as the same as those of the prototype structure for the case in this paper. As a consequence, the substructure model specimen could exhibit the same failure and crack distribution mode as the actual structure specimen, so as to achieve the experimental purpose of studying the static performance of the sidewall at the opening position. Figure 4 shows the dimensions and reinforcement details of the test specimen. The reinforcement ratio of each member in the station specimen was almost identical to that of the representative station, except for the differences in the bottom slab. In this study, the effect of the bottom slab on the bearing capacity of the entrance system was ignored. Hence, to ensure the restraint on the bottom ends of the columns and sidewalls

and to prevent bottom slab failure before substructure failure during the testing process, the thickness and reinforcement ratio of the bottom slab were strengthened. In addition, the steel ratios of the SRC column and SRC beams were approximately the same as those in the representative station.

**Table 1.** Parameters of the test specimen.

| Members | Section Size (mm) | $\rho_r$ (%) | $\rho_s$ (%) | $f_{cu,p}$ (MPa) | $f_{c,p}$ (MPa) | $f_{cu,c}$ (MPa) | $f_{c,c}$ (MPa) | $N_{ut}$ (kN) | $V_{ut}$ (kN) | $M_{ut}$ (kN·m) |
|---|---|---|---|---|---|---|---|---|---|---|
| SRC column | 400 × 350 | 3.74 | 5.90 | 55.6 | 35.8 | — | — | 7516.9 | — | 541.9 |
| SRC beam | 500 × 350 | 1.58 | 4.86 | 55.6 | 35.8 | — | — | — | 2203.7 | 2250.9 |
| Sidewall | 1500 × 350 | 1.07 | — | 60.1 | 38.6 | 49.6 | 32.1 | 15,535.1 | — | 365.3 |
| Top slab | 7300 × 400 | 1.28 | — | 60.1 | 38.6 | 40.9 | 27.3 | — | 3252.8 | 3417.5 |

Note: $\rho_r$ is the longitudinal reinforcement ratio of components; $\rho_s$ is the steel ratio; $f_{cu,p}$ is the cube compressive strength of precast concrete; $f_{cu,c}$ is the cube compressive strength of cast-in situ concrete; $f_{c,p}$ is the prism compressive strength of precast concrete; $f_{c,c}$ is the prism compressive strength of cast-in situ concrete; $N_{ut}$ is the theoretical value of the axial compressive bearing capacity of the member; $V_{ut}$ is the theoretical value of the shear resistant capacity of the member; $M_{ut}$ is the theoretical value of the ultimate flexural capacity of the member.

As shown in Figure 4, the cross-sectional dimensions of the scaled-down SRC column and beam were 400 × 350 mm and 500 × 350 mm, respectively. The sectional dimensions of the H-steel in the column were 250 mm × 200 mm × 12 mm × 14 mm (height × width × web thickness × flange thickness, respectively), and those of the H-steel in the beam were 350 mm × 200 mm × 9 mm × 14 mm, respectively. The cantilever H-steel beam integrated into the SRC column comprised sectional dimensions that were identical to those inside the beam. A bolting and welding mixed connection was utilized to connect the SRC column and SRC beam. This type of connection has already been successfully used by the authors in [36,37]. Thirty-four pieces of 14 mm diameter HRB400 longitudinal rebar were distributed around the cross-section of the SRC column, which had an approximate longitudinal reinforcement ratio of 3.74%. Grade HRB400 stirrups with a nominal diameter of 10 mm were arranged at a spacing of 100 mm along the column height. This arrangement accounted for a transverse reinforcement ratio of 0.96% in the columns. Simultaneously, the eighteen pieces of 14 mm diameter HRB400 longitudinal rebars were arranged around the SRC beam cross-section, and 10 mm diameter HRB400 stirrups were distributed at a longitudinal spacing of 150 mm along the beam span. The longitudinal reinforcement and stirrup ratios of the SRC beam were 1.58% and 0.30%, respectively.

The cross-sectional dimensions of the scaled-down single standard sidewall were 1500 mm × 350 mm, and the sectional dimensions of the scaled-down top slap were 7300 mm × 400 mm. The total height of the sidewall was 3300 mm, and the horizontal distance between the cantilevered and the fixed ends of the top slab was 2350 mm. As illustrated in Figure 4, grade HRB400 deformed 18 mm-diameter rebars with 150 mm spacing in the longitudinal direction were uniformly placed in the sidewall and the top slab to function as longitudinal reinforcement. The longitudinal reinforcement ratios of the sidewalls and top slab resulting from this configuration were 1.07% and 1.28%, respectively. Furthermore, the outer-side reinforcement of the sidewalls (i.e., tensile reinforcement) was arranged double-layered within a vertical range of 1500 mm from the bottom slab. This design can be attributed to the flexural moment at the bottom end of the sidewalls being significantly less than that at the top end under actual vertical loads (Figure 3). This could be utilized to guarantee that the expected flexural failure is controlled in the top region instead of the bottom end of the sidewall. Grade C50 concrete was used as the precast concrete of the entrance frame, sidewalls, and slabs. Grade C40 concrete was used for the cast-in situ concrete of the beam–column connection region, sidewalls, and top slab. In addition, the contacting surfaces of both the semi-precast top slab and sidewalls were roughened based on the recommendations in the concrete design specification [38] to transfer shearing force.

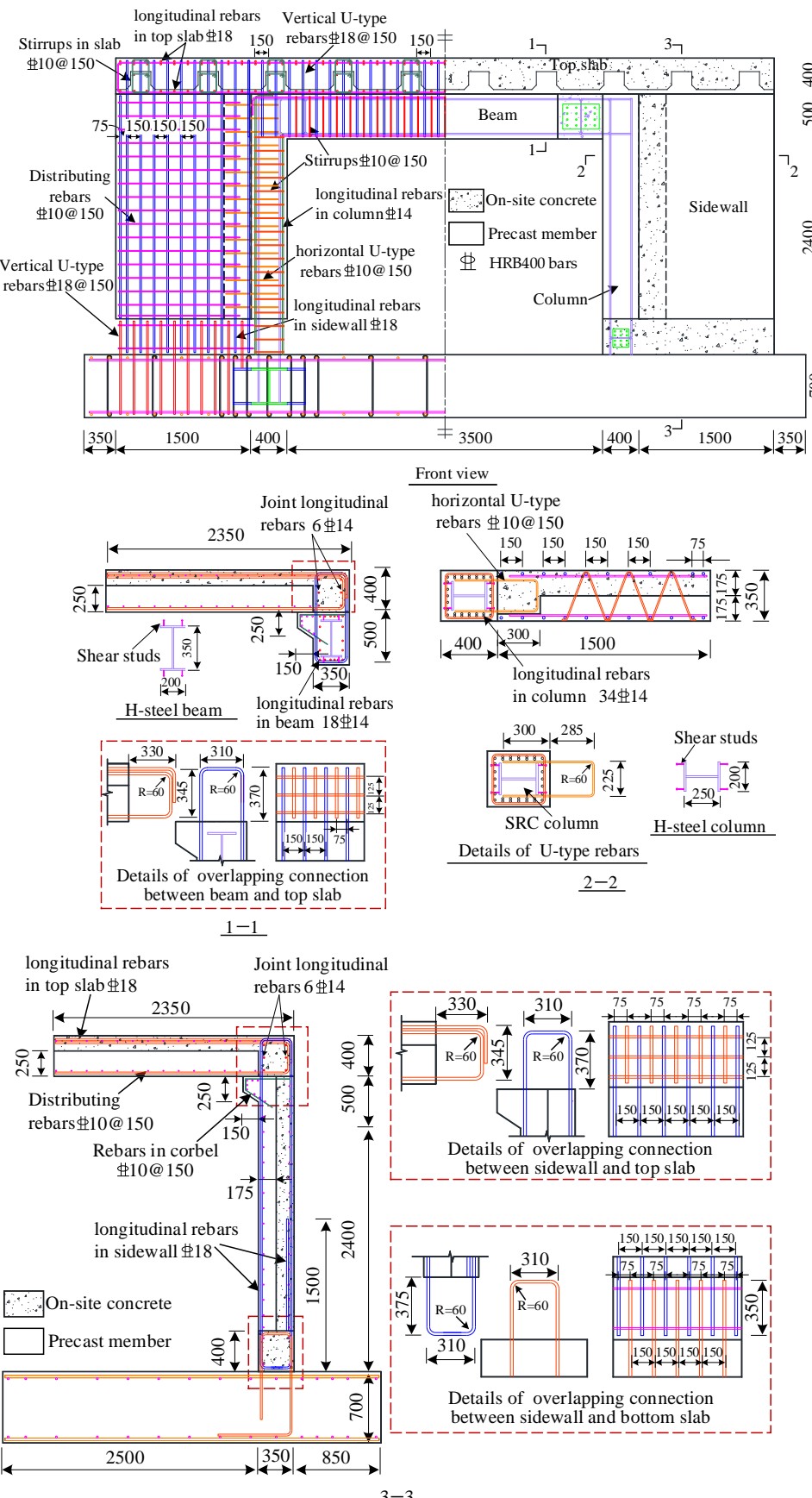

**Figure 4.** Dimensions and reinforcement details of the test specimen (units: mm).

Detailed U-type reinforcement overlapping connections are shown in Figure 4, in which it is observed that the design parameters of the top slab–sidewall connection and top slab–frame connections are similar. The total length of the horizontal and vertical overlaps of the adjacent U-type rebar was 655 mm. The center-to-center distance between adjacent U-type rebars in the longitudinal direction was 75 mm. In general, the beam–column knee joints with U-type reinforcement overlap have been commonly used in above-ground frame structures. Numerous experimental studies have verified the behavior of the knee joints and provided a basis for the current design methodology. As discussed in previous studies [39,40], joints with U-type reinforcements exhibited better mechanical behavior than those with standard 90 degree hook details. The developed knee joints for the top slab were designed based on the recommendations provided in the above literature, and the overlapping length of the lap splices met the Chinese code requirements [38]. For the connection between the sidewalls and the bottom slab, the bond strength evaluation of the U-type rebars was performed considering the bond strength of 90 degree hook-ended bars. Therefore, the U-type reinforcement overlapping connection can be considered as a general non-contact lap splice, and the connection parameters can be derived from both the current Chinese code [38] and the ACI building code [20]. The overlapping length of the non-contact splice was 350 mm, and the center-to-center distance between adjacent U-type rebars in the longitudinal direction was 75 mm. Simultaneously, the Chinese code recommendations [38] could be utilized to determine the overlapping length between the horizontal U-type rebars from the SRC columns and the distribution of the rebars along the sidewalls.

The construction process of the test specimens is illustrated in Figure 5. A partial precast top slab and sidewalls were prefabricated in a factory. For convenience, the bottom slab was manufactured directly at the designated test position. The H-steels of the entrance frame were installed in their designed positions before the concrete of the base plate was poured. The semi-sidewalls were then connected to a steel frame when the bottom slab was cured. Subsequently, the precast concrete encasing the steel frame and the on-site concrete of the sidewalls were poured in batches. Finally, the precast top slab was connected to the assembled frame and sidewalls. The precast part of the concrete in the specimen was cured under indoor conditions, and the cast-in-place part of the concrete was cured under outdoor conditions. During the curing process, attention was paid to the maintenance of the concrete surface temperature, and no obvious cracking of the concrete surface occurred.

### 2.3. Materials

To determine the characteristic value of cubic concrete compressive strength, six cubic specimens with dimensions of 150 mm × 150 mm × 150 mm were made for each batch of concrete. After curing for 28 days under identical conditions as the specimens, compression tests were performed on these cube specimens to determine the real concrete strength according to the standard GB 50204 [41], and the averaged results were calculated. The differences between the maximum and average results and between the minimum and average results of the cubic compressive strength of the six cubic specimens did not exceed 15% of the average result. According to the relevant provisions in standard GB 50204 [41], the average value could be taken as the cube compressive strength characteristic strength of this group of specimens. Based on the recommendations in standard GB 50010 [38], the prism compressive strength of concrete can be determined by the characteristic value of its cube compressive strength, which can be further used to calculate the theoretical bearing capacity of each member in the substructure. Both the cubic compressive strength of concrete ($f_{cu,p}, f_{cu,c}$) and the prism compressive strength of concrete ($f_{c,p}, f_{c,c}$) are shown in Table 1. The reinforcement and steel plate coupons were made from the corresponding batches of specimens. Based on the recommendations of standard GB/T 228.1 [42], coupon tensile tests were conducted on three coupons of each steel component and rebar to measure their material properties, and the averaged test results are listed in Table 2.

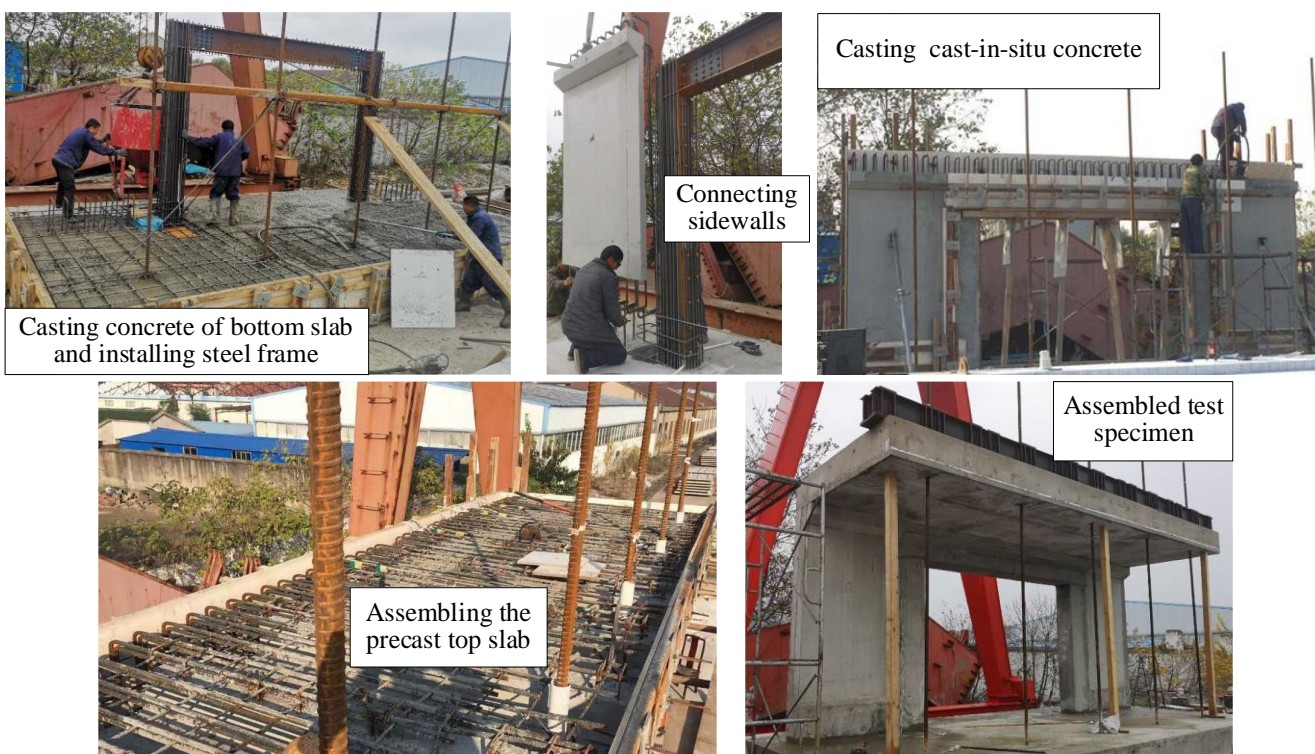

**Figure 5.** Construction process of the test specimen.

**Table 2.** Material properties of steel components.

| Elements | $t$ or $d$ (mm) | $f_y$ (MPa) | $f_u$ (MPa) | $E_s$ (GPa) | $\mathcal{E}_y$ (μ$\mathcal{E}$) |
|---|---|---|---|---|---|
| Flange of H-steel | 14 | 410.5 | 556.6 | 202 | 2032 |
| Web of H-steel | 12 | 407.6 | 551.2 | 201 | 2028 |
| | 9 | 411.5 | 552.0 | 201 | 2047 |
| | 18 | 425.0 | 640.2 | 201 | 2114 |
| Reinforcement | 14 | 440.1 | 630.3 | 201 | 2189 |
| | 10 | 455.0 | 550.0 | 201 | 2263 |

Note: $t$ is the thickness of steel components; $d$ is the diameter of reinforcements; $f_y$ is the yield strength of steel components and reinforcements; $f_u$ is the ultimate strength of steel components and reinforcements; $E_s$ is the modulus of elasticity; $\mathcal{E}_y$ is the yield strain of steel components and reinforcements.

### 2.4. Bearing Capacity of Components

The ultimate bearing capacity of each component can be utilized as a reference for formulating the loading protocol. The out-of-plane flexural capacity and axial compressive bearing capacity of the SRC columns were calculated according to the calculation assumptions in standards JGJ 138 [43] and EN 1994-1-1 [44]. The out-of-plane flexural capacity and axial compressive bearing capacity of the precast sidewalls were determined based on the recommendations of EN 1992-1-1 [18] and GB 50010 [38]. In addition, standards EN 1992-1-1 [18] and GB 50010 [38] can be used to evaluate the ultimate flexural capacity and shear resistance capacity of the precast top slab. The ultimate bearing capacity of each component in the substructure was calculated by using the measured strength values of steel shown in Table 2 and the calculated prism compressive strength of concrete shown in Table 1. The theoretical calculations of the bearing capacity of each member in the substructure are presented in Table 1. As observed, the flexural resistance of the top slab was higher than the total flexural resistance of the sidewalls and SRC columns. Furthermore, the calculated values of the shear resistance of the top slab indicated that bending failure of the laminated slab occurred before shear failure. Therefore, the typical flexural damage in the sidewalls was expected to be the primary failure mode of the test specimen.

*2.5. Test Setup and Loading Protocol*

A general view of the test setup is shown in Figure 6. Five high-strength finishing rolling rebars with a nominal diameter of 40 mm were used to connect the hydraulic jacks and top slab. When the concrete of the bottom slab was poured, the bottom ends of these high-strength rebars were vertically embedded in the bottom slab. Five hydraulic jacks, each with a maximum capacity of 1000 kN, were networked together to enable a single operator to apply a uniformly distributed load through the loading steel beam arranged along the longitudinal length of the top slab. The horizontal distance between the loading point and the center of the sidewall was 1935 mm, as shown in Figure 6a. The hydraulic fluid for these jacks was supplied by a power steering pump for controlling the total test loads by setting the working pressures. To measure the induced force, compression load cells were placed between each hydraulic jack and the loading steel beam. The loading process was initiated for a monotonous static downward load, with a total load increment of 44 kN, corresponding to a bending moment increment of 85 kN·m at the end of the top slab. A load-holding period of 5 min was maintained to observe the cracks, record the phenomena, and test the data. The displacement and strain information of the specimen during the loading process can be monitored by the field data acquisition instrument, as shown in Figure 6b. The arrangement of the measuring points of the specimen is reported in detail in Section 2.6. For safety, the test was terminated when all the longitudinal reinforcements in the tension zone of the sidewalls entered the yield stage, and the specimen had not yet reached the final failure.

*2.6. Test Instrumentation*

The response quantities of interest consisted of the crack patterns, overall deformation, and strain distribution of the sidewalls, entrance frame, and top slab. A concrete automatic digital crack width tester was used to monitor crack development from the initiation of cracks. Figure 6a illustrates the distribution of linear variable deformation transducers (LVDTs). Three LVDTs with a capacity of 300 mm (V1, V2, and V3) were installed at the bottom of the loading end of the top slab to measure the vertical deformation. Nine LVDTs (numbered H1–H9) with a capacity of 100 mm were installed horizontally at the backside of the sidewalls and entrance frame to respectively monitor their out-of-plane displacements.

Figure 7 presents the arrangement of the strain gauges for each member of the specimen. To measure the tensile strain distribution along the longitudinal direction of the sidewall, 10 general-purpose strain gauges (i.e., S1 to S10) were arranged longitudinally on the tensile reinforcement along cross-section 4 at a vertical distance of 800 mm from the beam bottom, as shown in Figure 7a. Simultaneously, 13 strain gauges (i.e., T1 to T13) were attached to the upper-layer reinforcement at the top slab end section to monitor the longitudinal strain distribution between the entrance opening and sidewalls, as shown in Figure 7b. To measure the stress state of the SRC column, strain gauges CR1 and CR2 were arranged on the compressive and tensile reinforcements of the column, respectively, and strain gauges F1 to F4 were placed on the column flange at cross-section 4.

To monitor the stress distribution around the opening, eight strain rosettes (each consisting of three 60 mm gauges) were attached to the concrete surface on the rear end of the frame (i.e., the tension side) at the locations shown in Figure 7a. Strain rosettes CW1 to CW4 were arranged vertically from the bottom of the beam to the top of the slab, well aligned with the corner of the opening. Strain rosettes CW5 to CW8 were arranged vertically from the bottom of the beam to the top of the slab, which were aligned to the opening centroid. Moreover, six strain rosettes (i.e., SW1 to SW6), each with three general gauges, were used to measure the stress distribution between the supports and midspan of the steel beam, and their arrangements are illustrated in Figure 7a. In addition, the strain development of U-type rebars in the connection zone was measured for better understanding of the joint behavior. As shown in Figure 7c, strain gauges TJ1 to TJ3 and ST1 to ST3 were used to monitor the U-type rebars in the assembly joint between the

sidewall and top slab, while strain gauges SB1 to SB6 and BB1 to BB4 were used to measure the U-type rebars in the assembly joint between the sidewall and bottom slab.

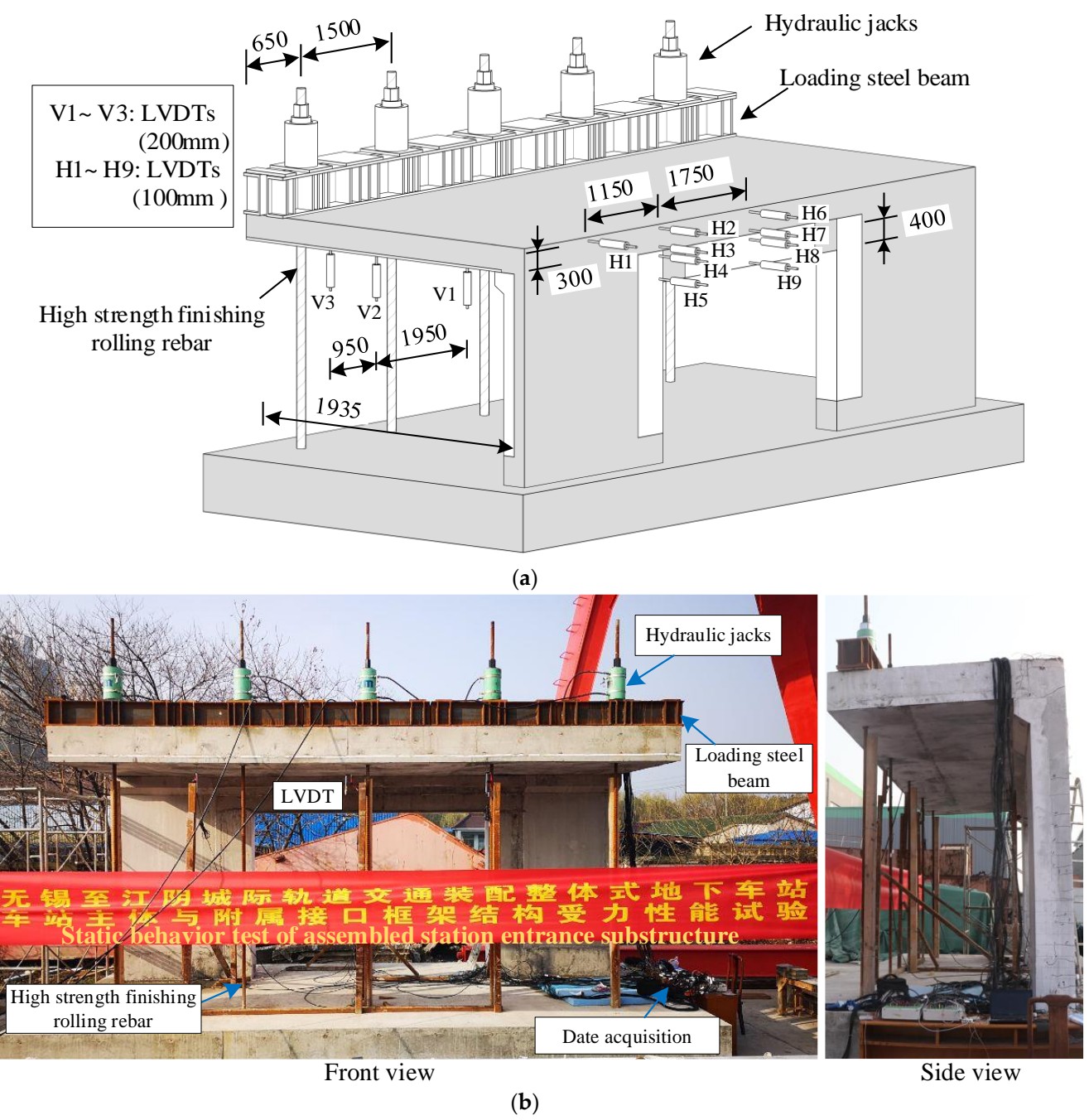

**Figure 6.** General overview of a test setup (units: mm): (**a**) schematic of the test setup and (**b**) photograph of the test setup.

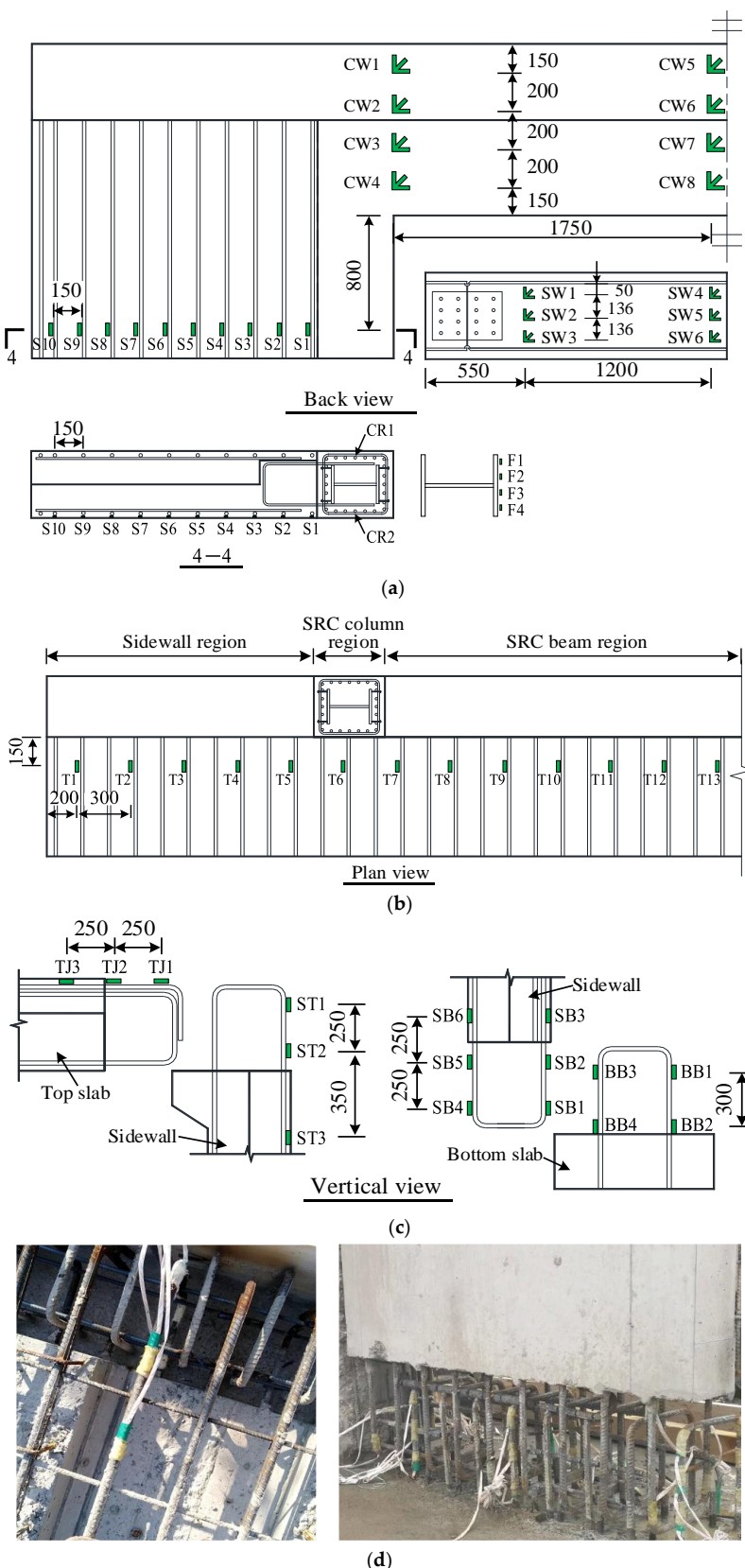

**Figure 7.** Arrangement of strain gauges (units: mm): (**a**) strain-gauge layout in sidewall and entrance frame, (**b**) strain-gauge layout in the top slab, (**c**) strain-gauge layout in the U-type rebar overlapping connections, and (**d**) photograph of strain gauge arrangement at U-type rebar overlapping connections.

## 3. Experimental Results

### 3.1. Overall Behavior and Crack Pattern

The total applied load measured by the compression load cell versus the horizontal and vertical displacements measured by the LVDTs is shown in Figure 8. Here, the vertical displacement at the loading end of the top slab (TSVD) was measured using LVDTs V1–V3. It can be seen that the incremental difference in the TSVD along the longitudinal direction of the station was negligible. Furthermore, the out-of-plane horizontal displacements at the top of the sidewall (STHD) and the top of the entrance frame (ETHD) were monitored using LVDTs H1 and H6, respectively, and the measurements indicated an insignificant difference between the two displacement increments. As the formal loading was terminated when all tensile rebars of the sidewalls entered the yield stage, the applied load increased almost linearly, with extremely limited inelastic deformations. The total applied loads and displacements corresponding to each stage of the test specimen during the loading process are shown in Figure 8.

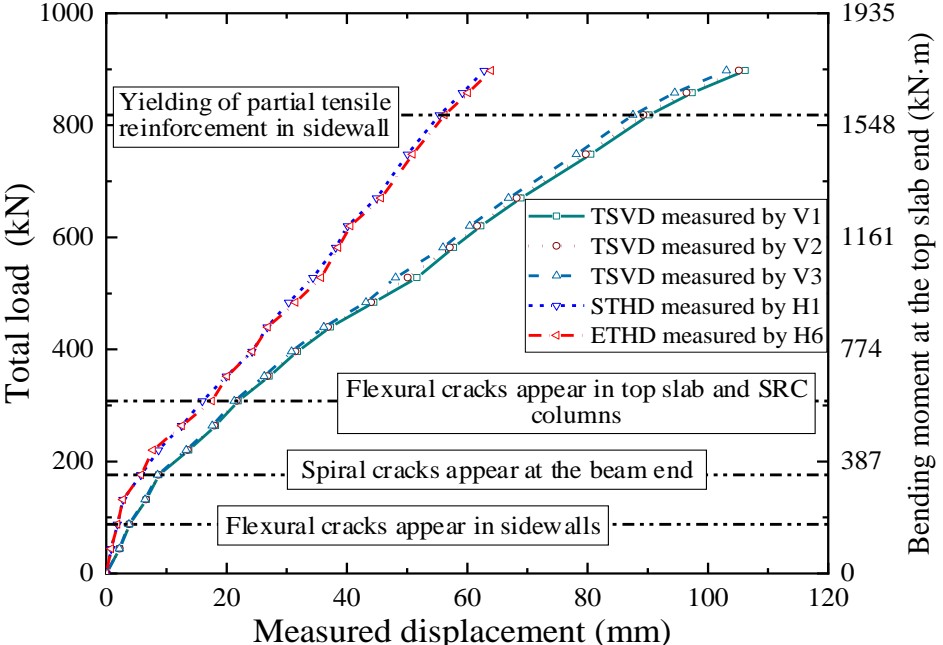

**Figure 8.** Load displacement curves for the specimen.

The crack pattern of the test specimen at the yield stage of all tensile reinforcements in the sidewalls is illustrated in Figure 9. As shown in the back view, the first flexural crack was observed on the tension side of the sidewalls when the applied load reached 88 kN, corresponding to a bending moment of 170.3 kN·m at the end of the top slab. The load level at this stage was approximately $0.1P_y$. Here, $P_y$ is the maximum load corresponding to the load at which all tensile reinforcements in the sidewalls yielded. Subsequently, the number of flexural cracks along the vertical direction of the sidewalls gradually increased, and the crack width increased as the applied load increased. As the applied load increased, the horizontal cracks appearing on the sidewalls further increased in the SRC column section. This further indicates the effectiveness of the connection between the SRC column and sidewalls in transferring stress.

When the applied load of 176 kN was measured, corresponding to a bending moment of 340.6 kN·m at the end of the top slab, diagonal cracks were initiated at both ends of the SRC beam corresponding to the corners of the opening. The load level at this stage was approximately $0.2P_y$. With a further increase in the applied load, these diagonal cracks gradually propagated to the fixed end of the top slab. The angle between the diagonal cracks and the beam axis ranged from 35° to 55°. As shown in the front view in Figure 9,

diagonal cracks could also be observed at the beam ends corresponding to the corners of the opening. Interestingly, the propagation direction of the front diagonal crack was opposite to that of the diagonal cracks on the rear side. It is therefore deduced that the diagonal cracks appearing on the front and back of the beam ends may be credited to the torsional deformation of the SRC beam.

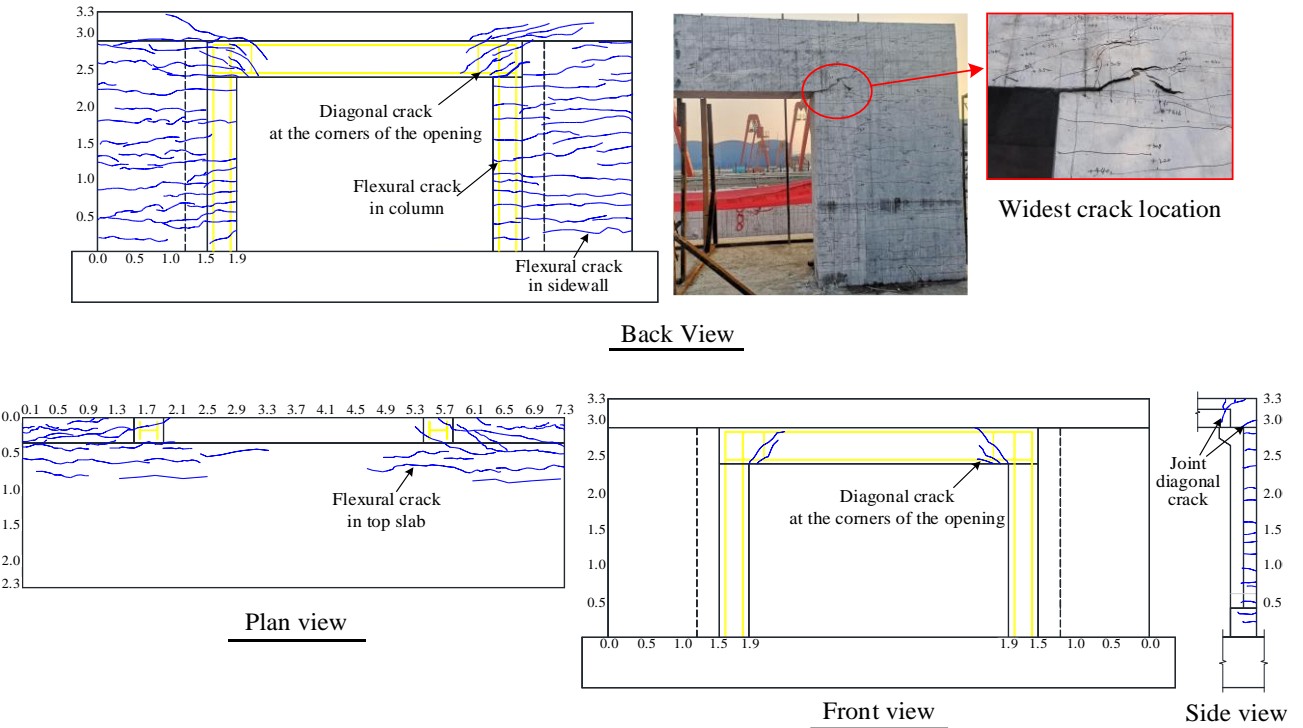

**Figure 9.** Crack patterns of the specimen (units: m).

When the horizontal cracks in the sidewalls spread over the entire column section, flexural cracks were also observed in the top slab at an applied load of 308 kN (approximately $0.35P_y$). As shown in the plan view in Figure 9, the propagation of flexural cracks was concentrated at the fixed end of the slab within the range of the sidewalls, and only a limited number of cracks grew in the opening region throughout the test. Upon further increasing the applied load to 396 kN (approximately $0.45P_y$), a few diagonal cracks initially appeared at the joint between the sidewalls and top slab, as shown in the side view in Figure 9. Interestingly, the joint diagonal cracks did not propagate to the core region throughout the loading process, and the maximum crack width was 0.3 mm. In addition, no splitting cracks occurred at the location of the U-type rebar, implying that no bond slippage occurred between the U-type rebar and concrete throughout the test.

When the applied load increased to 818.5 kN, corresponding to a bending moment of 1583.8 kN·m at the end of the top slab, partial tensile reinforcement in sidewalls entered the yield stage. The load level at this stage was approximately $0.9P_y$. In addition, a horizontal crack with a width of 2.5 mm and length of 150 mm appeared in the right column section that intersected with the bottom of the SRC beam, as shown in the back view in Figure 9. Subsequently, this crack propagated obliquely to the top of the sidewall at an angle of approximately 45° and continued to cause the concrete cover to spall at the top of the sidewall. As the load increased further, this crack became the widest crack, finally attaining its maximum width (3.5 mm). As the reinforcement in the sidewall began to yield, the load increased nonlinearly until it reached a maximum of 897.5 kN ($P_y$) when the test was terminated.

*3.2. Crack Width Development*

Figure 10 shows a comparison of the crack width development at the opening corners, SRC columns, top slab end, and sidewalls of the specimen. As mentioned in [17,33], the presence of cut-outs in traditional concrete walls encourages cracks to appear first at the corners of the opening. However, owing to the entrance frame setting in the specimen, cracks in the sidewalls appeared before those at the corners of the opening. At the initial loading stage, the width of the flexural crack in the sidewall was the largest compared to other members. When the total applied load was increased to 582 kN, the crack width at the corners of the opening gradually exceeded that of the sidewalls. The extremely rigorous crack width control requirement in underground structures is meant for the expectation of providing excellent resistance to the penetration of water and chemicals. Notably, the development of the crack width at the corners of the opening was more abrupt than that in other members. Therefore, more attention must be paid to the reason for the formation and width control of cracks at the opening corners. For the top slab and sidewalls, only the maximum crack widths during each monotonic loading process are shown in Figure 10. In actuality, the width of the horizontal flexural cracks in both the sidewall and top slab gradually increased from the opening to the outside edges, thereby directly demonstrating the lag effect of the entrance frame on the stress development of longitudinal rebars in adjacent members.

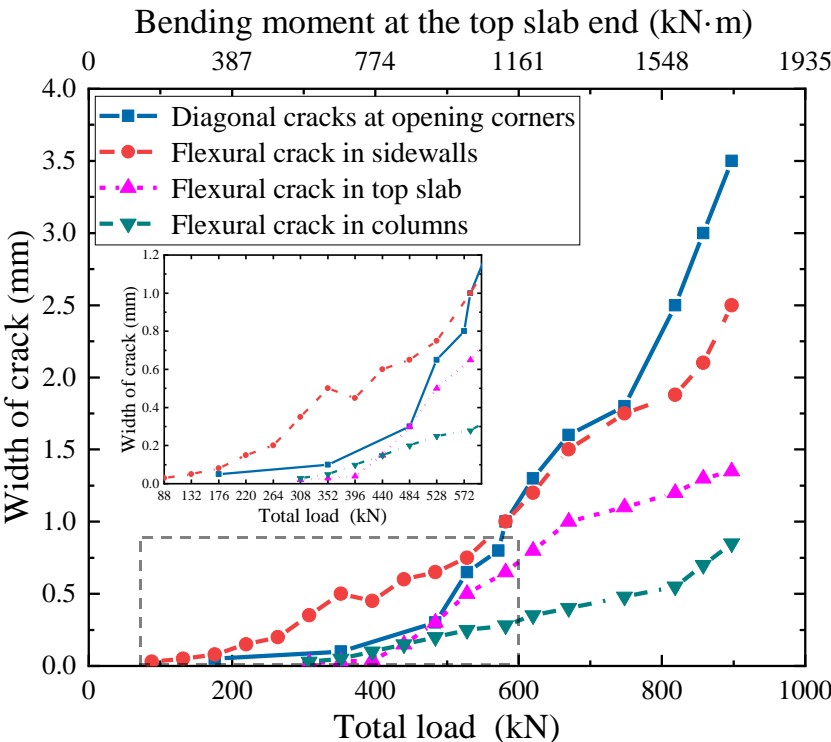

**Figure 10.** Load–crack width curves of the specimen.

*3.3. Strains of the U-Type Reinforcement*

Figure 11a illustrates the relationships between the applied load and the strains of the U-type reinforcement in the connection between the sidewalls and the bottom slab. Gauges SB6 and SB3 were attached to the longitudinal reinforcement outside the overlapping region of the U-type rebar. Their arrangement was utilized to directly realize the development of the longitudinal strain in the sidewall. Furthermore, at each load level, the longitudinal strain value of the U-type reinforcement decreased gradually from the sidewall to the core region of the joint. This was attributed primarily to the presence of bond stress between the overlapping rebars and surrounding concrete. In such cases, the longitudinal strain of the overlapping U-type rebar maintained elasticity throughout the test.

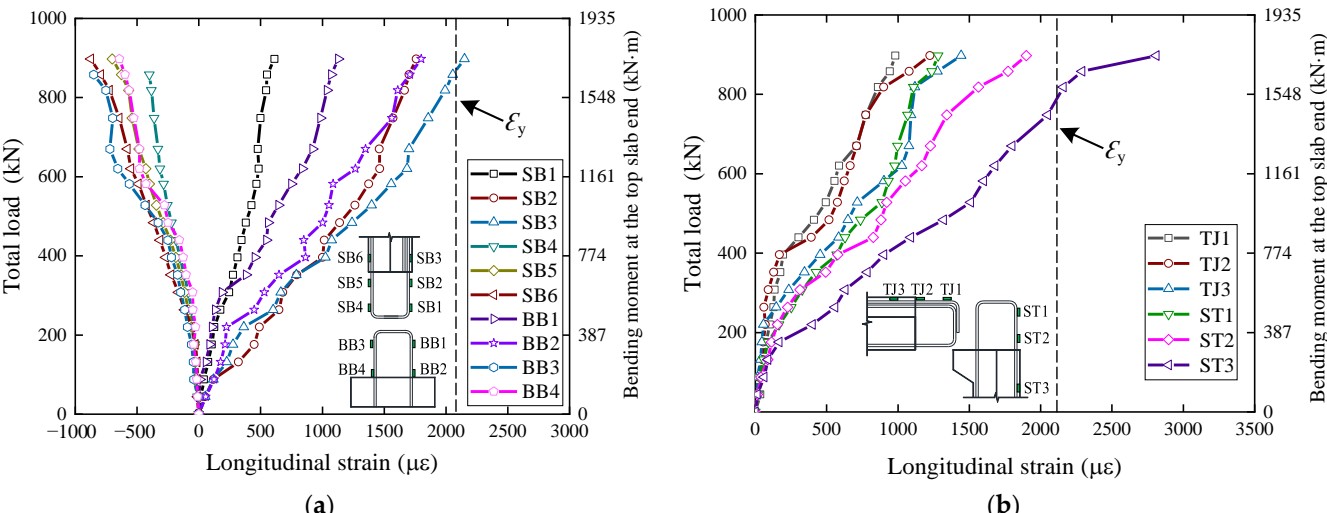

**Figure 11.** Strains of the U-type reinforcement at the joint: (**a**) connection of sidewalls and bottom slab and (**b**) connection of sidewalls and top slab.

Figure 11b further shows the development of the tensile strain of U-type rebars in the connection between the sidewalls and top slab. Similar to the lap joint at the bottom of the sidewall, the transferring stress attained between the overlapping U-type rebars in the connection was also primarily determined by the bond stress around rebars. Therefore, the minimum longitudinal reinforcement strain appeared at the ends of the U-type rebars extending from the respective members, where the gauges TJ1 and ST1 were located. As a result of the larger flexural resistance at the end of the top slab, only the tensile reinforcement at the top end of the sidewall entered the yield stage, and during the testing process, no overlapping U-type rebars were yielded. Moreover, as discussed in Section 3.1, no bond-splitting cracks occurred in the top and bottom joints of the sidewall. It was therefore deduced that there was no bond stress degradation between the U-type reinforcement and the concrete in the proposed connection type, and the manner of overlapping U-type reinforcements could effectively guarantee the stress transfer between the sidewalls and top and bottom slabs.

*3.4. Strains of the Sidewall Longitudinal Reinforcement*

The longitudinal reinforcement strain distribution along the sidewall cross-section is shown in Figure 12, which can be used to further analyze the effects of the entrance frame on the stress distribution within the sidewall. As observed, the longitudinal reinforcement in the edge of the sidewall, further from the entrance frame, attained yield strength first during the testing process. Subsequently, the tensile reinforcement in the sidewall region close to the frame began to gradually yield. It could be seen that the presence of the entrance frame had a lag effect on the stress development of the longitudinal reinforcement in its adjacent local regions. This further demonstrated that the developed entrance frame could effectively prevent stress concentration on both sides of the opening and could facilitate crack width control in the sidewalls.

Figure 13 demonstrates the development of longitudinal strains in the SRC column under different load levels. The distribution of the longitudinal steel strains along the H-steel flange width is illustrated in Figure 13a, and the applied load versus longitudinal reinforcement strain curves are given in Figure 13b. Evidently, the SRC column and sidewalls jointly resisted the out-of-plane flexural moment from the top slab end. The distance between the neutral axis and the centerline of the column was approximately 55 mm. Both the longitudinal reinforcement and the H-steel in the column remained in the elastic stage throughout the test. This phenomenon could be attributed to only a limited sidewall section size being selected in the substructure design. However, the longitudinal

section of the sidewall is much larger than that of the SRC column in actual subway stations. In such cases, the flexural resistance of the column has a negligible effect on the bearing capacity of the sidewalls. The experimental results showed that the yielding of all tensile rebars in the sidewalls occurred earlier than concrete crushing in the compression zone. Therefore, it is deduced that although the sidewalls were also subjected to vertical loads, the failure was primarily controlled by the out-of-plane bending moment.

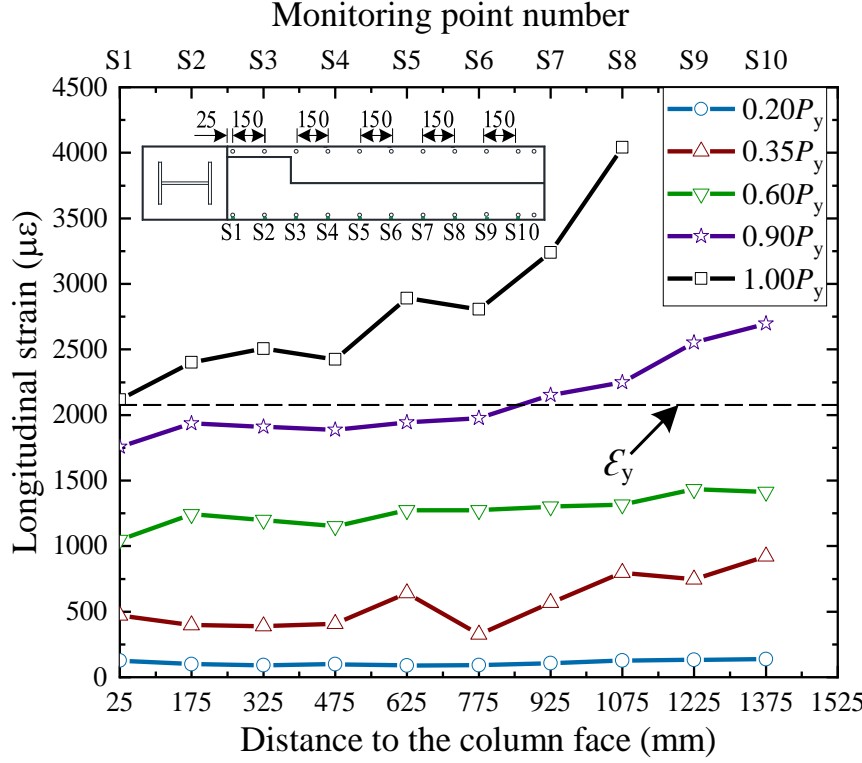

**Figure 12.** Strain distribution along the sidewall cross-section.

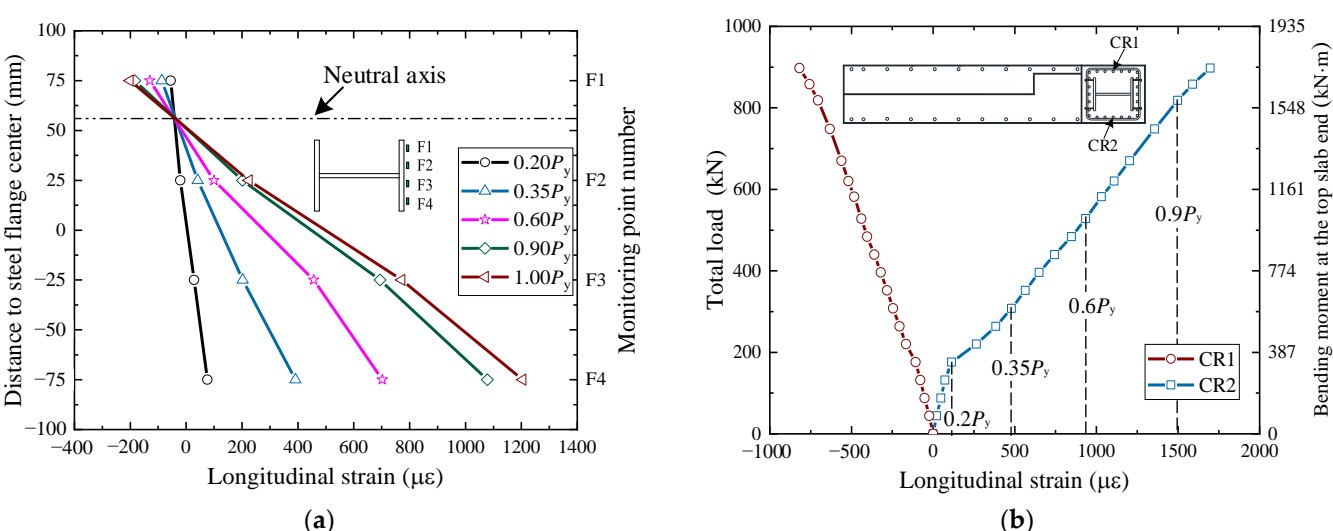

**Figure 13.** Strains development in the SRC column: (**a**) typical strain distribution along the flange and (**b**) load–strain curves of the longitudinal rebars.

*3.5. Strains of the Top Slab Longitudinal Reinforcement*

Figure 14 shows the strain distribution of the longitudinal reinforcement at the end of the top slab from the outer edge of the sidewall to the opening. Note that the longitudinal

rebars at the slab end still maintained elasticity, even when all tensile reinforcements in the sidewalls yielded. Furthermore, the distribution of the reinforcement strain values at the slab end generally exhibited a decreasing trend from the sidewall to the opening, and it only marginally increased in the range of the SRC column. The strain distribution further illustrated the transmission mechanism of the bending moment of the slab end to the sidewall and entrance frame. It was therefore deduced that the out-of-plane moment from the slab end was primarily carried by the sidewalls, and that an extremely limited moment load was transferred to the SRC beam above the opening. However, the limited bending moment borne by an SRC beam must receive attention. This may be the primary reason for torsional deformation of the frame beam. In addition, the out-of-plane bending moment carried by the beam was further transmitted to the SRC column through the beam–column connection. As a result of this process, diagonal cracks may be generated at the corners of the opening, that is, on the front and rear of the beam ends.

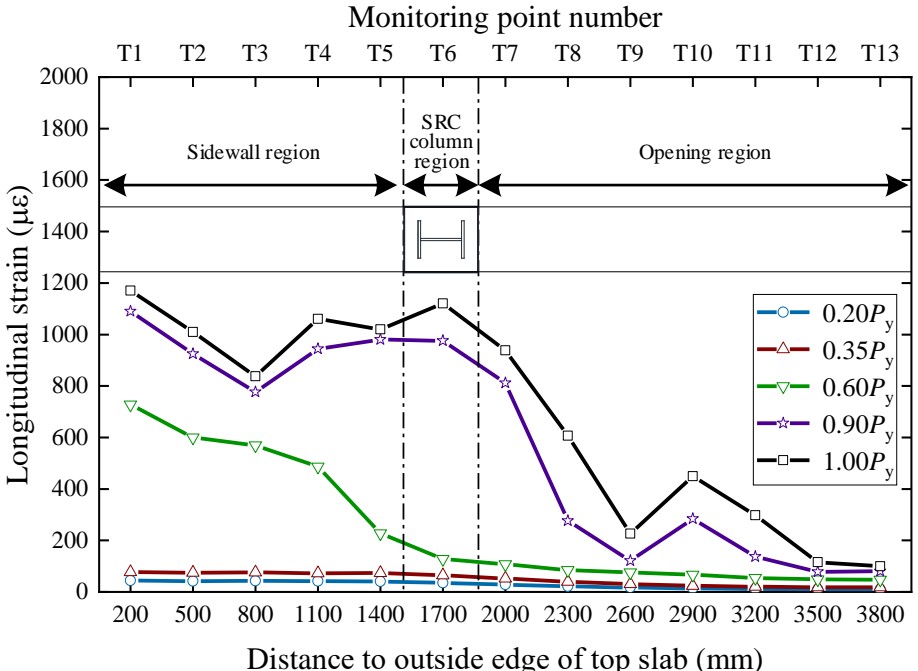

**Figure 14.** Strain distribution of the longitudinal reinforcement at the top slab end.

### 3.6. Stressed State of the SRC Beam

To measure the out-of-plane torsional deformation of the SRC beam during the testing process, a series of horizontal LVDTs (H4, H5, H8, and H9) was installed at the rear side of the beam, as shown in Figure 6a. Here, LVDTs H4 and H5 were almost aligned to the SRC beam end section, and LVDTs H8 and H9 to the midspan section of the SRC beam. The torsion angle ($\theta$) of the midspan section of the SRC beam can be calculated as follows:

$$\theta = \arctan\frac{\delta_{H8} - \delta_{H9}}{400} - \arctan\frac{\delta_{H4} - \delta_{H5}}{400} \tag{1}$$

where $\delta_{Hi}$ is the reading of LVDT numbers H4, H5, H8, and H9 during the test. During the final loading stage, the experimental value of $\theta$ was evaluated to be 0.007 rad. The captured $\theta$ value directly demonstrated an extremely slight torsional deformation of the beam.

To further illustrate the stressed state of the SRC beam under different load levels, the principal stress along the beam depth was monitored using strain rosettes attached to the concrete surface and the H-steel web at the locations shown in Figure 7a. In the previous stage when the diagonal cracks appeared, the principal stress of the concrete obtained from strain rosettes CW1 to CW8 is shown in Figure 15a. It can be seen that the maximum principal stress of the concrete at the corner region of the opening increased

continuously from the top of the slab to the bottom of the beam. This directly resulted in the different development of the diagonal crack width along the beam depth. In addition, the angle between the major principal stress direction and the beam axis is also shown in Figure 15a. The principal tensile stress direction at the beam end was at 46–66° to the beam axis. However, the angles between the principal tensile stress direction at the beam midspan section and beam axis were drastically reduced to 5–18°. Furthermore, the principal stress values of the beam midspan section were almost zero. Figure 15b further illustrates the variation in the principal stress of the H-steel between the midspan and supports of the beam at the last loading level, where the stress was measured from strain rosettes SW1 to SW6. Similar to the concrete stress case, the major principal stress of the H-steel decreased significantly from the end to the midspan of the beam, and the corresponding directions were also gradually oriented toward the beam axis.

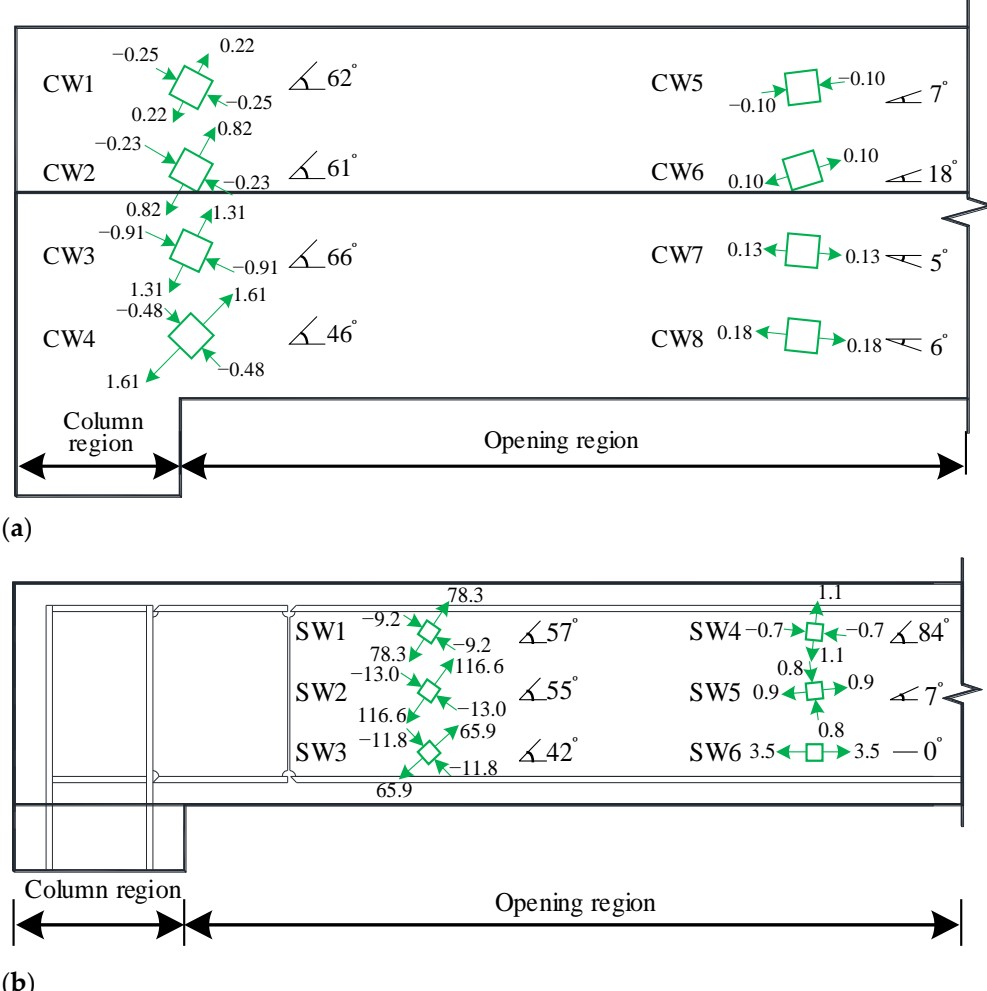

**Figure 15.** Stress distribution of the SRC beam (units: N/mm$^2$): (**a**) stress distribution of the concrete before cracking and (**b**) stress distribution of H-steel in the last loading stage.

The reason for this significant difference can be attributed to the variation in the torsional moment along the beam length. As described in Section 3.5, after the bending moment was transmitted from the top slab to the SRC beam, the load could then be further transferred in the torsional moment from the beam to the columns through beam-to-column connections. Therefore, the torsional moment was primarily concentrated at the beam ends, which could be considered the primary cause of stress development at the beam ends. Owing to the constraint of the column on the torsional deformation of the beam ends, the principal stress direction was not at an angle of 45° or 135° to the beam axis. It

had a negligible effect on stress development owing to the lower torsional moment at the midspan location. Therefore, it was deduced that the development of diagonal cracks at the open corners was primarily dominated by the torsional moment at the beam ends.

## 4. Finite Element Analysis

To further examine the effect of the proposed entrance frame on the static behavior of adjacent members under vertical loading and provide brief design criteria for the frame, ABAQUS software was utilized to construct a three-dimensional (3D) finite element (FE) model of the substructure. Nonlinear inelastic analysis was conducted using the ABAQUS/implicit static general procedure. The FE model is illustrated in Figure 16.

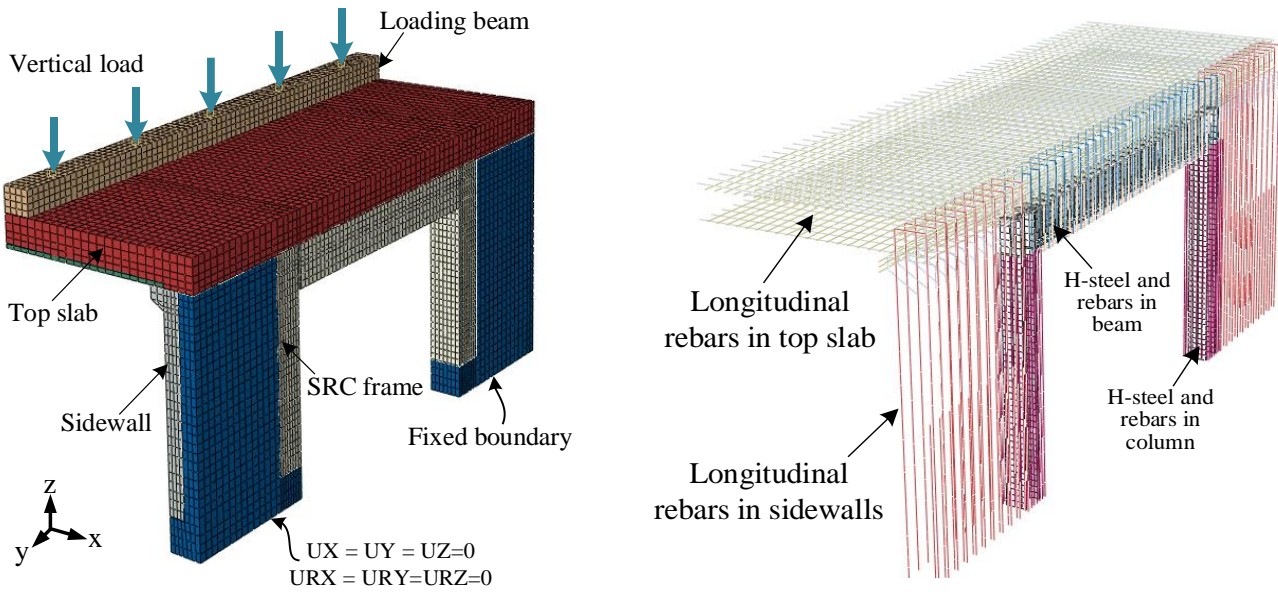

**Figure 16.** Meshed model and the loading methods.

### 4.1. FE Model

In the model, the cross-sectional parameters and reinforcement details of each member in the model were identical to those of the test specimen. The concrete and H-steel were discretized using the eight-node linear 3D solid elements with hourglass control (C3D8R). All reinforcing bars were modeled by the three-dimensional two-node linear truss elements (T3D2). To ensure accurate prediction of the static behavior and deformability of the specimen at a reasonable computational cost, a sensitivity study was conducted by discretizing continuous solid elements in different regions using a variety of nominal element sizes. The meshing of the model is also shown in Figure 16. Here, the nominal concrete element sizes in the thickness direction of the sidewalls and top slab were 50 mm and 100 mm, respectively. In the longitudinal direction of the station, the nominal concrete member sizes of the sidewalls and top slab were both 100 mm. The concrete and H-steel of SRC columns were discretized in three dimensions using an element size of 50 mm. The three-dimensional mesh size of concrete and H-beam in the longitudinal 500 mm region at both ends of the SRC beam was 50 m, while the mesh size of the remaining regions was 100 mm. In addition, 100 mm truss elements were utilized to discretize the reinforcement. To simulate the interactions of the interfaces between cast-in-place concrete and precast concrete in the laminated slabs, the "hard contact" and the "penalty–friction" algorithms from the default interaction library of ABAQUS were adopted [45]. The friction coefficient between the interfaces of different concrete layers was set as 0.6, which was determined based on the results of a previous study [46]. Embedded technology was used to couple the degrees of freedom between concrete and steel rebar elements and between concrete and H-steel elements, since no bond slippage occurred during the testing process.

The boundaries and loading conditions of the model were applied based on the test results. Because of the constant restraint of the bottom slab to the sidewalls and entrance frame and the effectiveness of the bottom joints of the sidewall throughout the test, fixed boundary conditions were uniformly formulated to simulate the constraints of the bottom slab on the sidewalls and frame. Here, the degrees of freedom in the X-, Y-, and Z-directions at the bottom ends of the sidewalls and frame were restricted, respectively. To reproduce a uniformly distributed load at the top slab end, a loading beam model with a flexural stiffness similar to that of the experimental steel beam was developed. The modeled beam was tied at the end of the top slab using the constraint feature in ABAQUS [47]. A vertical load was directly applied at the top of the modeled beam and controlled by the loading protocol used in the test.

The rebar and H-steel in the model were treated as the elastic–plastic materials, and the material characteristics in compression and tension were similar. The multiple linear kinematic hardening model [48] can be used to describe the mechanical behavior of steel. The parameters of the steel model were determined based on the material test results, as summarized in Table 2. The stress–strain curves of the steel are depicted in Figure 17a. The axial compressive and tensile plasticity damage model [49] can be used to analyze the mechanical performance of concrete. The critical parameters of the model were obtained from the material test results, as listed in Table 1. The compressive and tensile stress–strain curves of the concrete were drawn based on the method proposed in GB 50010 [38], as shown in Figure 17b,c. In the concrete damage plasticity (CDP) model, the dilation angle is 30°, the eccentricity is 0.1, the viscosity parameter is 0.005, and the ratio between the concrete strength in terms of biaxial and uniaxial strength is 1.16. The concrete damage factors of different components in the CDP model can be determined according to the recommendations provided in the ABAQUS standard user's manual [47], as shown in Equations (2) and (3).

$$D_c = 1 - \frac{\sigma_c E_c^{-1}}{\varepsilon_c^{in}(1 - \eta_c) + \sigma_c E_c^{-1}} \tag{2}$$

$$D_t = 1 - \frac{\sigma_t E_c^{-1}}{\varepsilon_t^{in}(1 - \eta_t) + \sigma_t E_c^{-1}} \tag{3}$$

where $D_c$ and $D_t$ are the tensile and compressive damage factors of materials in the CDP model, respectively; $\eta_c$ is the ratio of compressive damage plastic strain to inelastic strain, which is 0.7; $\eta_t$ is the ratio of tensile damage plastic strain to inelastic strain, which is 0.1; $\varepsilon_t^{in}$ is inelastic tensile strain; $\varepsilon_c^{in}$ is inelastic compressive strain; $E_c$ is the elastic modulus of concrete; and $\sigma_c$ and $\sigma_t$ represent the compressive stress and tensile stress of concrete, respectively.

### 4.2. Validation of the FE Model

The developed FE model was validated by comparing the numerical results with the test results in terms of the load–deformation responses and crack patterns. A comparison of the load–deformation diagrams obtained from the FE analysis and test is presented in Figure 18. Although the displacements calculated by the FE model were marginally less than the experimental results after the cracking of the top slab concrete, the predicted displacements under the $P_y$ load were basically consistent with the experimental results. Generally, the load–deformation diagrams obtained from the FE analysis resembled well the experimental results.

The maximum principal inelastic strain patterns of the concrete materials from the FE analysis could be used to simulate the distribution of cracks in the specimen. Figure 19 displays the principal inelastic strain distributions of the concrete materials under the $P_y$ load. As shown in Figure 19, the most critical plastic damage occurred at the SRC beam ends and the adjacent top slab and SRC columns, coinciding with the diagonal crack-concentration regions of the specimens during the test. The principal strain on the sidewalls gradually decreased from the outside edge to the entrance frame. This trend was

in favorable agreement with the development of the longitudinal reinforcement strain and concrete crack width along the sidewall cross-section. The comparison confirmed that the developed FE model could accurately show the development of cracks around the opening.

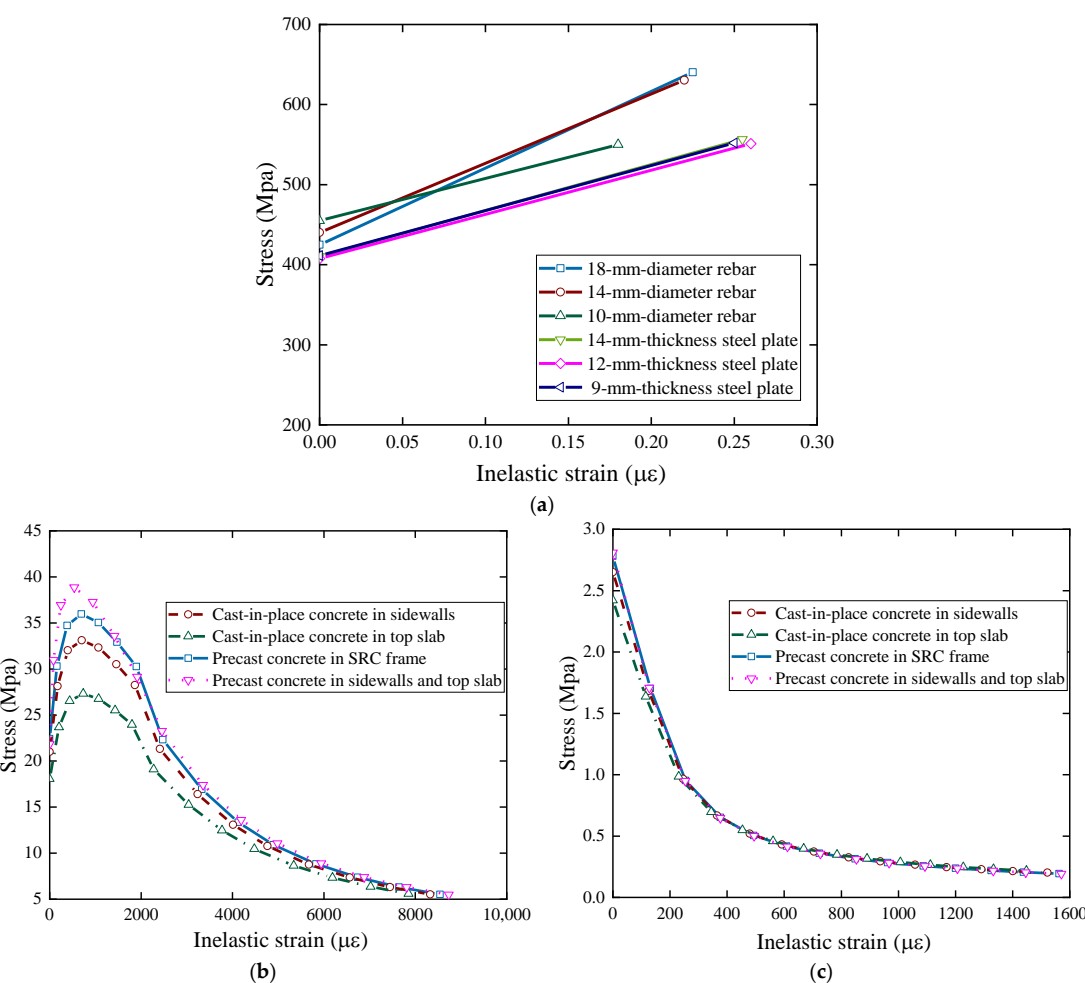

**Figure 17.** Stress–strain curves of steel and concrete: (**a**) steel constitutive curve, (**b**) concrete compressive stress–strain curve, and (**c**) concrete tensile stress–strain curve.

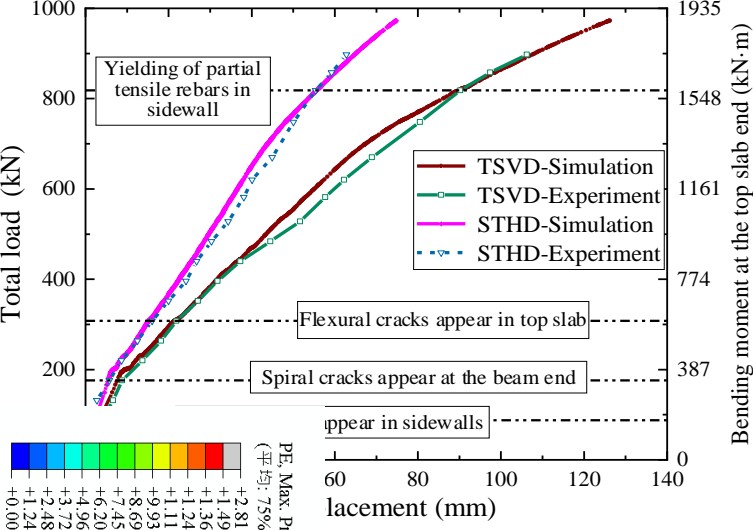

**Figure 18.** Comparison of load–deformation responses.

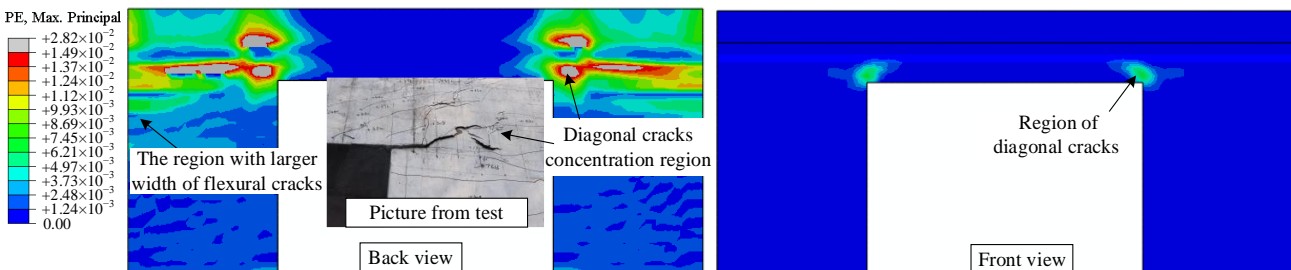

**Figure 19.** Maximum principal inelastic strain patterns from FE analysis.

*4.3. Parametric Study*

A parametric study was conducted using the validated FE model to further investigate the effects of the entrance frame with different configurations on the development of cracks around the opening. The parameters were the steel ratio of the SRC frame and cross-sectional height of the column and beam. Table 3 summarizes the details of all cases in this parametric study. For convenience, these cases are denoted as E-X. Case E-1 employed a validated FE model with a configuration similar to that of the test specimen, which could be treated as the baseline configuration for the other cases. Compared to E-1, the steel ratio of the SRC beam in E-2 was zero, and that of the SRC column in E-3 was zero as well. Both cases were designed to investigate the effects of the steel ratio of the entrance frame. In addition, the cross-sectional height of the SRC beam increased to 800 mm in Case E-4, and the cross-sectional width of the SRC column increased to 550 mm in Case E-5. An increase in the cross-sectional heights of the SRC beam and column directly improved the bearing capacity and out-of-plane stiffness. For Cases E-4 and E-5, although the cross-sectional heights of the SRC column or beam were varied, their longitudinal reinforcement ratios and steel ratios for all frame sections were still similar to those in Case EJ-1.

**Table 3.** Details and results of different cases in the parametric study.

| Name | SRC Beam Section | | | | SRC Column Section | | | | $M_y$ (kN·m) | $T_y$ (kN·m) | $k_1$ (%) | $k_y$ (%) |
|---|---|---|---|---|---|---|---|---|---|---|---|---|
| | Height (mm) | Width (mm) | $\rho_s$ (%) | $\rho_r$ (%) | Height (mm) | Width (mm) | $\rho_s$ (%) | $\rho_r$ (%) | | | | |
| E-1 | 500 | 350 | 4.86 | 1.58 | 400 | 350 | 5.90 | 3.74 | 1807.7 | 264.5 | 21.2 | 29.3 |
| E-2 | 500 | 350 | 0.00 | 1.58 | 400 | 350 | 5.90 | 3.74 | 1694.6 | 174.8 | 21.4 | 20.6 |
| E-3 | 500 | 350 | 4.86 | 1.58 | 400 | 350 | 0.00 | 3.74 | 1504.7 | 172.2 | 21.5 | 22.9 |
| E-4 | 800 | 350 | 4.86 | 1.58 | 400 | 350 | 5.90 | 3.74 | 1498.0 | 261.3 | 26.3 | 34.9 |
| E-5 | 500 | 350 | 4.86 | 1.58 | 400 | 550 | 5.90 | 3.74 | 3798.6 | 799.8 | 26.8 | 42.1 |

*4.4. Effect on Beam End Torques*

Table 3 shows the data when all the tensile reinforcements in the sidewalls yielded, that is, reached the $P_y$ load, the bending moment at the top slab end ($M_y$), and the torsional moment at the beam end ($T_y$) in each case. It can be seen that the varied steel ratios and cross-sectional heights of the SRC column and beam contributed to the yield resistance of specimens. Nevertheless, as discussed in Section 3.4, the dimensions of the sidewall in the actual project were significantly larger than those of the entrance frame. Therefore, the variations in the bearing capacity and out-of-plane stiffness of the frame contributed insignificantly to the overall mechanical performance of the sidewall systems. A proportionality coefficient ($k$) can be utilized to further illustrate the effect of configuration parameters on the proportion of the load carried by the beams above the opening. $k$ can be defined as the ratio of the summation of the torques ($T$) at both ends of the beam to the corresponding bending moment ($M$) at the top slab end, i.e., it can be calculated as $2T/M$. Here, the coefficient $k_1$ was obtained when the concrete in the FE model was not cracked, and the coefficient $k_y$ was obtained under the $P_y$ load. It should be noted that $T$ was obtained by

integrating points over the entire height of the section from the bottom of the SRC beam to the top of the roof in the FE analysis.

In the elastic stage, the coefficient $k_1$ values varied from 21.2% to 26.8%, as shown in Table 3. The difference between the coefficient $k_1$ values of Cases E-1 to E-3 with different steel ratios were negligible, which indicates that the variation in the steel ratio of the entrance frame had negligible effects on the load distribution coefficient of the SRC beams and its adjacent members. For Cases E-4 and E-5, the coefficient $k_1$ values were 24.1% and 26.4% higher than those of Case E-1, respectively. This implied that both the improvement in the torsional stiffness of the SRC beam and in the out-of-plane flexural stiffness of the SRC column could significantly increase $T$ at the beam ends in the elastic stage. In general, the steel ratio exhibited a marginal contribution to the frame stiffness. This resulted in a negligible effect on the SRC frame sharing of the out-of-plane moments from the top slab end. Consequently, the variation in the steel ratio of the SRC frame contributed limitedly to $T$ at the beam ends under identical loading levels.

As can be seen from Table 3, the coefficient $k_y$ values ranged from 20.6% to 42.1% when all tensile reinforcements in the sidewalls of the respective models entered the yield stage. For E-1, E-3 E-4, and E-5, the coefficient $k_y$ values were 38.2%, 6.5%, 32.6%, and 57.1% higher than those of $k_1$, respectively. However, the coefficient $k_y$ of E-2 was 3.7% lower than $k_1$. The change in coefficients implies a redistribution of the internal force between the SRC beams and its adjacent members, which can be attributed to the constant variation in the torsional stiffness of the SRC beam and in the flexural stiffness of the vertical members during the loading process due to concrete cracking and reinforcement yielding in the members. Owing to the rapid development of cracks at the beam ends of E-2, the torsional stiffness degradation of its beam was more pronounced than that of other cases. Therefore, its proportionality coefficient presented an overall decreasing trend with increasing load. For other cases, the increase in $k_y$ could be explained by the development of flexural cracks in the sidewalls exacerbating the degradation of its flexural stiffness, and changes of the configuration parameters restraining the cracks of beam ends. This also indicates that the steel ratio of the SRC beam had a significant effect on the control of crack development at the beam ends.

*4.5. Effect on Inelastic Strain Distribution of Concrete*

Figure 20 further shows the maximum principal inelastic strain distributions of the concrete materials under the $P_y$ load for Cases E-2 to E-5. Similar to E-1 (Figure 19), the maximum plastic damage of concrete materials in E-2 was also concentrated at the SRC beam ends, and the damage region gradually expanded to the top slab and sidewalls. Note that the plastic damage area of E-2 was significantly greater than that of E-1, which can be explained by the lower torsional capacity of the normal concrete beam than that of the SRC beam. In contrast to E-1 and E-2, the location of the initial plastic damage at the beam ends of E-3 was almost aligned to the central section of the SRC column. This phenomenon can be attributed to the absence of H-steel in the column, which significantly reduced the resistance of the column to the torsional moment transmitted from the beam end. It is therefore deduced that the variations in the flexural resistance of the column had a significant effect on the development of the cracks at the open corners.

Cases E-4 and E-5 effectively limited the development of concrete inelastic strain at the beam ends. For E-4, although the increment in the cross-sectional heights of the SRC beam directly increased the torsional moment shared by the beam, the torsional bearing capacity improved simultaneously as well. In addition, the higher beam section effectively reduced the torsional shear stress transmitted to the column, which could prevent the column concrete from entering the plasticity prematurely. For E-5, the increment in the cross-sectional height of the SRC column significantly improved the out-of-plane flexural capacity and its restraint on the torsional deformation of the beam ends. Therefore, this method was effective in preventing column concrete from entering plasticity and limiting further propagation of the beam end torsional shear stress in the top slab. The aforementioned

results prove that both the improvement in the torsional bearing capacity of the beam and out-of-plane flexural capacity of the column positively influenced the control of crack development at the open corners.

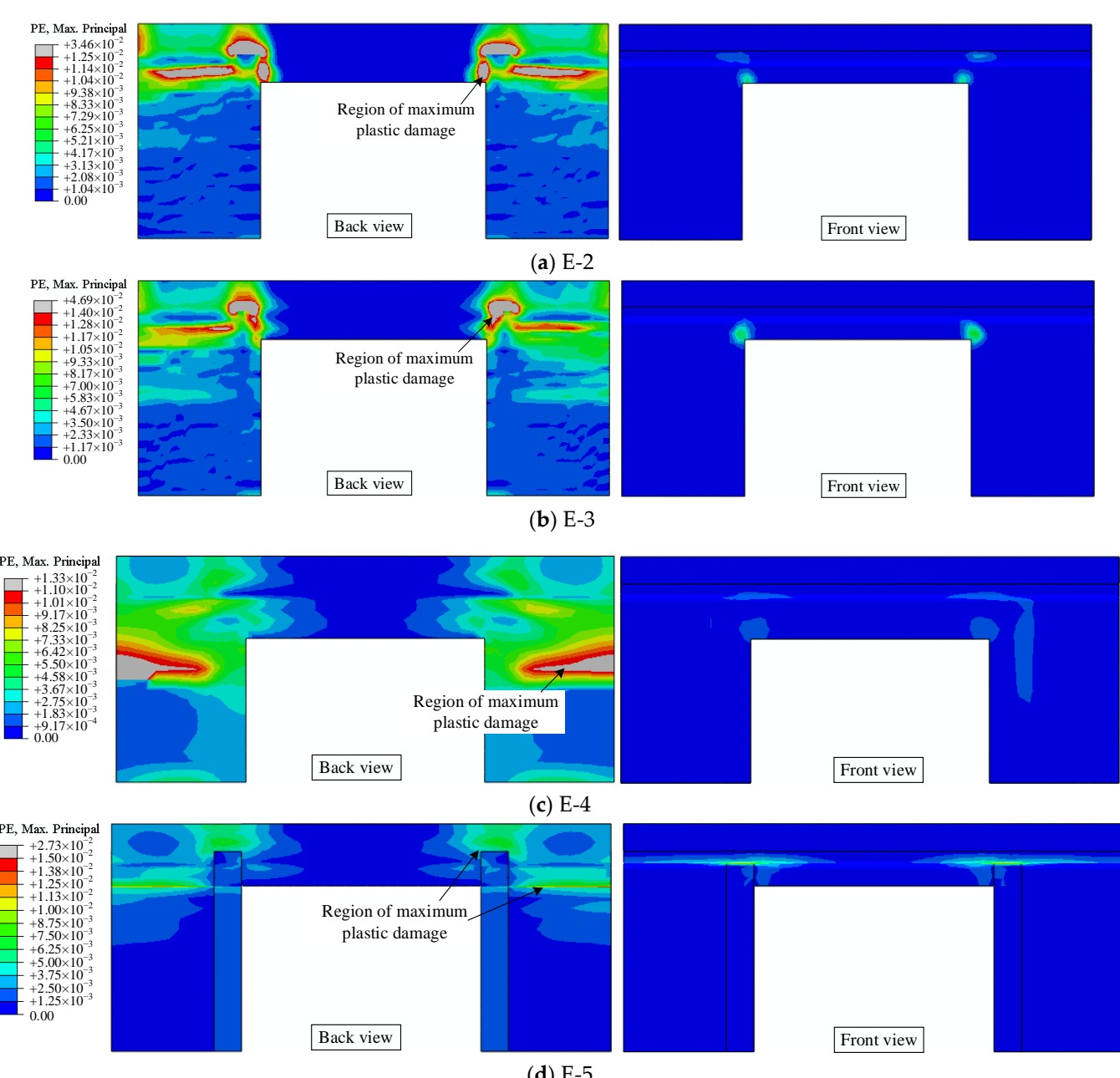

**Figure 20.** Maximum principal inelastic strain patterns of the concrete for Cases E-2 to E-5.

## 5. Conclusions

In this study, a novel type of precast entrance frame was proposed to compensate for the effects of entrance openings on the localized bearing capacity of the continuous sidewalls and prevent stress concentrations around the openings. To further investigate the effect of the developed entrance frame on the mechanical behavior of its adjacent sidewalls, a monotonic static test and finite element analysis were performed on a 1/2 scale station entrance substructure. The overall static behavior, cracking behavior, and stress distribution of the substructure were reported. The main findings can be summarized based on the experimental and numerical results as follows:

(1)	The U-type rebar overlapping connection between the developed frame and adjacent members exhibited satisfactory mechanical performance throughout the test, implying that this connection type could effectively guarantee stress transfer between precast members.

(2)	The developed frame influenced the stress distribution of the adjacent members. Furthermore, it could effectively prevent stress concentration in the adjacent sidewall region and facilitate crack-width control in the sidewalls.

(3)	The out-of-plane moment transmitted from the top slab end to the SRC beam was concentrated at the beam ends in the form of torsional moment. This directly resulted in a stress concentration at the corner region of the opening, which could be regarded as the primary cause of the serious development of cracks in this region.

(4)	Parametric studies were performed to investigate the influence of the steel ratio of the SRC frame and cross-sectional dimensions in the frame on the proportion of the load carried by the beam above the opening. The results showed that both the improvement in the torsional stiffness of the SRC beam and the out-of-plane flexural stiffness of the SRC column could significantly increase the torsional moment at the beam ends under identical rooftop vertical loads.

(5)	Parametric studies also investigated the influence mechanism of the developed frame with different configuration parameters on the crack development around the opening. The results showed that the improvement in the torsional bearing capacity of the SRC beam as well as the out-of-plane flexural capacity of the SRC column could contribute positively to the control of crack development around the opening.

Note that the number of test samples in this study was relatively limited due to the limitations of testing costs. In addition, due to the limitation of available experimental operation space, the test specimen was designed as a scale substructure. To address this research gap, further finite element studies can be carried out based on the validated FE model to simulate the mechanical behavior of the overall station structure with an entrance opening, and to investigate the effects of other variables not investigated in the present study.

**Author Contributions:** Conceptualization, Z.M. and D.G.; methodology, S.F. and D.G.; software, S.F.; validation, S.F., Z.M. and D.G.; riting—original draft preparation, S.F.; writ-ing—review and editing, D.G.; visualization, S.F.; supervision, Z.M. and Z.G.; project administration, Z.G.; funding acquisition, Z.M. and Z.G. All authors have read and agreed to the published version of the manuscript.

**Funding:** This research is supported by the National Natural Science Foundation of China (52278154) and the Fundamental Research Funds for the Central Universities (2242021R10031).

**Data Availability Statement:** Some or all data, models, or codes generated or used during this study are available from the corresponding author by request.

**Acknowledgments:** Thanks for the supported by the National Natural Science Foundation of China (52278154) and the Fundamental Research Funds for the Central Universities (2242021R10031).

**Conflicts of Interest:** The authors declare no conflict of interest.

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
