# Peer review of "Effect of Entrance Frame on Crack Development around Prefabricated Subway Station Openings"

_buildings, doi:10.3390/buildings13041032_

Round 1
Reviewer 1 Report
This article presents research leading to the development of a prefabricated reinforced concrete frame (SRC) that is sufficiently resistant to the stresses concentrated around openings in the side walls of subway station entrances.
It is very well done, but needs to provide better justification for the need for research and strengthening of its cause.
You could present in a table the previous studies carried out on this topic and the problems identified, results, recommendations, methods used, etc.
Figure 6: What information is displayed in the red band? if it's important to the test, translate. If not, replace the photo with one that doesn't have the banner.
Reviewer 2 Report
In this manuscript, a monotonic static test was performed on a 1/2 scale station entrance substructure, including the proposed entrance frame and the adjacent top slab, bottom slab, and sidewalls. A three–dimensional (3D) finite-element (FE) model was developed to simulate the substructure’s static behavior. However, the research work is not very revealing and the results are not soundly innovative. The following comments should be considered:
1) The abstract is too long, the conclusion needs to be condensed, and there is no sense of innovation.
2) There should be more experimental studies on the mechanical properties of walls under out-of-plane loads, and relevant studies should be supplemented in the first part.
3) From the point of view of the design of the test, the test can completely enter the failure stage, why not enter the overall yield or failure to obtain the full curve?
4) The cloud figure of finite element analysis should be compared with the test fracture figure.
5) Fig.18 shows that since the finite element results are in good agreement with the test results, can the peak point or even the decline curve of the finite element analysis be given?
6) Because the loading of this test is relatively simple, the forces on different sections of the specimens are the same, and the mechanical performance is different from the actual engineering situation. It is suggested to supplement the analysis under the combined forces of in-plane and out-of-plane.
Reviewer 3 Report
The present work reports experimental and numerical analyses of pre-fabricated subway station openings. The following issues were identified in this manuscript:
1. The last paragraph of Section 1 should summarize the research gap and highlight the main original contributions (issues that were never investigated in previous works) provided by the present study.
2. The authors must prove that the scaled-down specimens represent actual case scenarios based on scalability theory.
3. Number of specimens and standard methods used in concrete specimens testing were not provided in Section 2.3.
4. Number of specimens and standard methods used in steel specimens testing were not provided in Section 2.3.
5. Compaction procedures used to cast all types of concrete specimens were not explained.
6. Concrete curing conditions (temperature and moisture conditions) were not reported.
7. Did you show average or characteristic values of compressive strength of concrete in Table 1?
8. Did you use average strength of concrete cubes of 150 mm to determine the theoretical values presented in Table 1? Did all design codes mentioned in Section 2.4 consider concrete strength as the average strength of concrete cubes of 150 mm?
9. It is necessary to clarify in Section 2.5 how the authors identified that all the longitudinal reinforcements in the tension zone of the sidewalls entered the yield stage. This section did not state if steel strain was measured or not.
10. Sketches of Figure 6 and Figure 7 must be complemented by pictures of actual specimens. For example, Figure 6b could also show a photograph of the back side of the actual specimen, in addition to photograph the front side. Figure 7 could also show some photographs of the strain gauges bonded to the different parts of the specimen.
11. In Section 4, the authors used the Concrete Damage Plasticity (CDP) for numerical models if concrete. Despite this, many input parameters of the CDP model were not mentioned in this paper, such as the ratio between the concrete strength in the biaxial and uniaxial strength, dilation angle, viscosity parameter, and eccentricity parameter. Did you use the same parameters for the concrete of the different members of specimen?
12. The authors did not describe the mesh convergence studies used to select the mesh size for each structural component of the numerical analyses.
13. How did you model concrete damage level in the numerical analyses?
14. Section 5 should highlight the novelty of the present study.
Round 2
Reviewer 2 Report
The author responded to the review comments reasonably and revised the manuscript. I believe the paper is appropriate for publication in Buildings.
Author Response
Thank you for your suggestions to help our manuscript quality has been further improved.Reviewer 3 Report
The authors solved some problems of the papers. However, the following issues were not corrected:
1. Response to comment #2 is not appropriate because the authors did not prove that the scaled-down specimens were able to represent actual case scenarios based on scalability theory. Scale model experiments must be based on the similarity principle of elasticity, considering appropriate changes in physical parameters and geometric sizes recommended in previous literature [e.g., SEDOV and VOLKOVETS, Similarity and dimensional methods in mechanics].
2. Response to comment #7 is not suitable because Table 1 did not indicate if the compressive strengths were provided in terms of average or characteristic values. In order to solve this issue, the authors should revise the difference between characteristic strength and average strength of concrete.
3. Response to comment #8 is not clear. The experiments were carried out on concrete cubes. In the response to comment #8, the authors mentioned the use of “prism compressive strength”. Did you use compressive strength of prisms to determine the theoretical values presented in Table 1? It is not clear in the text. If so, how did you obtain the prism concrete strength from the cube concrete strength, considering that the compressive strength of concrete cubes is certainly higher than the compressive strength of concrete prisms?
4. Did all design codes mentioned in Section 2.4 consider concrete strength as the average strength of concrete cubes of 150 mm? Or average strength of concrete prisms? Or characteristic strength of concrete cubes? Or characteristic strength of concrete prisms?
5. Response to comment #10 is incomplete. Figure 7d should clarify the designation and/or location of the strain gauges presented in this new image.
6. Response to comment #13 is incomplete. In order to clarify the methodology in the manuscript, the paper could show the Equations (1) and (2) provided in the response to reviewer, in addition to the selected values of Æžc and Æžt, according to recommendations of relevant literature.
Author Response
"Please see the attachment."

Round 3
Reviewer 3 Report
The authors corrected the problems identified by this reviewer and the quality of the paper was improved.